# Mutant p53 drives clonal hematopoiesis through modulating epigenetic pathway

Sisi Chen[1,2], Qiang Wang [3], Hao Yu[2], Maegan L. Capitano[4], Sasidhar Vemula[2], Sarah C. Nabinger[2], Rui Gao[2], Chonghua Yao[2,5], Michihiro Kobayashi[2], Zhuangzhuang Geng[3], Aidan Fahey[2], Danielle Henley[2], Stephen Z. Liu[6], Sergio Barajas[1], Wenjie Cai[1], Eric R. Wolf [2], Baskar Ramdas[2], Zhigang Cai[2], Hongyu Gao[6], Na Luo [7], Yang Sun [7], Terrence N. Wong[8], Daniel C. Link[8], Yunlong Liu[6], H. Scott Boswell[9], Lindsey D. Mayo[2], Gang Huang[10], Reuben Kapur[2], Mervin C. Yoder[2], Hal E. Broxmeyer[4], Zhonghua Gao[3]* & Yan Liu[1,2]*

Clonal hematopoiesis of indeterminate potential (CHIP) increases with age and is associated with increased risks of hematological malignancies. While *TP53* mutations have been identified in CHIP, the molecular mechanisms by which mutant p53 promotes hematopoietic stem and progenitor cell (HSPC) expansion are largely unknown. Here we discover that mutant p53 confers a competitive advantage to HSPCs following transplantation and promotes HSPC expansion after radiation-induced stress. Mechanistically, mutant p53 interacts with EZH2 and enhances its association with the chromatin, thereby increasing the levels of H3K27me3 in genes regulating HSPC self-renewal and differentiation. Furthermore, genetic and pharmacological inhibition of EZH2 decreases the repopulating potential of p53 mutant HSPCs. Thus, we uncover an epigenetic mechanism by which mutant p53 drives clonal hematopoiesis. Our work will likely establish epigenetic regulator EZH2 as a novel therapeutic target for preventing CHIP progression and treating hematological malignancies with *TP53* mutations.

[1] Department of Biochemistry and Molecular Biology, Indiana University, Indianapolis, IN 46202, USA. [2] Department of Pediatrics, Herman B Wells Center for Pediatric Research, Indiana University, Indianapolis, IN 46202, USA. [3] Department of Biochemistry and Molecular Biology, the Cancer Institute, College of Medicine, Pennsylvania State University, Hershey, PA 17033, USA. [4] Department of Microbiology and Immunology, Indiana University, Indianapolis, IN 46202, USA. [5] Shanghai Municipal Hospital of Traditional Chinese Medicine, Shanghai University of Traditional Chinese Medicine, Shanghai, China. [6] Department of Medical Genetics, Indiana University, Indianapolis, IN 46202, USA. [7] Department of Ophthalmology, Indiana University, Indianapolis, IN 46202, USA. [8] Siteman Cancer Center, Washington University, St. Louis, MO 63110, USA. [9] Department of Medicine, Indiana University, Indianapolis, IN 46202, USA. [10] Division of Pathology and Experimental Hematology and Cancer Biology, Cincinnati Children's Hospital Medical Center, Cincinnati, OH, USA. *email: zgao1@pennstatehealth.psu.edu; liu219@iu.edu

Clonal hematopoiesis of indeterminate potential (CHIP), also known as age-related clonal hematopoiesis (ARCH), occurs when a single mutant hematopoietic stem and progenitor cell (HSPC) contributes to a significant clonal proportion of mature blood lineages during aging[1–3]. CHIP is common in aged healthy individuals and associated with increased risks of hematological neoplasms, including myelodysplastic syndromes (MDS) and acute myeloid leukemia (AML)[4–8]. CHIP is also associated with increased all-cause mortality and risk of cardio-metabolic disease[4–6,9]. While these findings suggest that mutations identified in CHIP likely drive disease development, mechanisms by which these mutations promote HSPC expansion are largely unknown[4–9].

Most individuals with CHIP carry hematological malignancy-associated mutations, including *DNMT3A, TET2, ASXL1, JAK2*, and *TP53*[4–6]. The *TP53* gene, which encodes the tumor suppressor protein p53, ranks in the top five among genes that were mutated in CHIP[4–6,10–12]. p53 bears the usual hallmarks of a transcription factor and regulates a large number of genes in response to a variety of cellular insults, including oncogene activation, DNA damage, and inflammation, to suppress tumorigenesis[13,14]. *TP53* mutations and deletions were found in approximately half of all human cancers, including hematological malignancies[13,14]. Recently, somatic *TP53* mutations were identified in CHIP[4–6]. *TP53* mutations were also commonly found in therapy-related CHIP[10,12]. Interestingly, some individuals with Li-Fraumeni syndrome (LFS), who carry germline *TP53* mutations, develop MDS and AML as they age[14,15]. Indeed, somatic *TP53* mutations are present in 10% of MDS and AML cases and in 30% of secondary MDS and AML patients arising after exposure to radiation or chemotherapy[2,16–19]. While *TP53* mutations are associated with adverse clinical outcomes in MDS and AML[2,16–19], how mutant p53 drives the pathogenesis of hematological malignancies are not fully understood.

We have been investigating the role of p53 in normal and malignant hematopoiesis. We discovered that wild-type (WT) p53 maintains hematopoietic stem cell (HSC) quiescence and identified Necdin as a p53 target gene that regulates DNA damage response (DDR) in HSCs[20,21]. We extended our research to mutant p53 to generate additional knowledge in order to develop therapeutic strategies that can enhance our abilities to prevent CHIP progression and treat hematological diseases. We discovered that mutant p53 enhances the repopulating potential of HSPCs[22]. While clinical studies suggest that expansion of HSPCs with *TP53* mutations predisposes the elderly to hematological neoplasms[4–6,10–12], the role of *TP53* mutations in CHIP progression remains elusive.

Polycomb group (PcG) proteins are epigenetic regulators that have been implicated in stem cell maintenance and cancer development[23–26]. Genetic and biochemical studies indicate that PcG proteins exist in at least two protein complexes, Polycomb repressive complex 2 (PRC2) and Polycomb repressive complex 1 (PRC1), that act in concert to initiate and maintain stable gene repression[23–26]. EZH2, a key component of PRC2 complex, catalyzes the trimethylation of lysine 27 of histone H3 (H3K27me3) in cells[25]. While EZH2 plays important roles in HSCs and MDS development[16,27,28], its regulation in HSPCs is not fully understood.

In the present study, we discovered that mutant p53 confers a competitive advantage to HSPCs following transplantation and promotes HSPC expansion after radiation-induced stress. Mechanistically, mutant p53 interacts with EZH2 and enhances its association with the chromatin, thereby increasing the levels of H3K27me3 in genes regulating HSPC self-renewal and differentiation. Thus, we have uncovered an epigenetic mechanism by which mutant p53 drives clonal hematopoiesis.

## Results

### *TP53* mutations identified in CHIP enhance HSPC functions.

*TP53* ranks in the top five among genes that were mutated in CHIP (Fig. 1a)[4–6,10–12]. Approximately 90% of somatic *TP53* mutations in CHIP are missense mutations in the DNA-binding domain (DBD) of the p53 protein (Fig. 1b)[4–6,10–12]. The most frequently mutated codon in p53 was 248, followed by codons 273, 220, and 175 (Fig. 1c). *TP53* mutation spectrums in CHIP are similar to hematological malignancies. Different mutant p53 proteins have been shown to exhibit distinct functions in promoting cancer initiation, progression, or metastasis[14]. To determine the impact of *TP53* mutations on HSPC functions, we introduced eight hot-spot *TP53* mutations identified in CHIP[4–6,10–12] (Fig. 1c), into WT mouse HSPCs using retrovirus-mediated transduction and performed in vitro and in vivo assays (Fig. 1d). Ectopic expression of some mutant p53, including p53$^{R248W}$, p53$^{R248Q}$, p53$^{R175H}$, p53$^{R273H}$, p53$^{C238Y}$, and p53$^{Y220C}$, enhanced the replating potential of WT HSPCs compared to control viruses (MIGR1) transduced cells (Fig. 1e).

p53$^{R248W}$, p53$^{R273H}$, and p53$^{Y220C}$ are hot-spot *TP53* mutations in CHIP, MDS, and AML and predict leukemia development[4–6,10–12,17–19]. These mutations have also been shown to gain oncogenic properties in mouse models of human cancer[14,29–31]. We introduced p53$^{R248W}$, p53$^{R273H}$, or p53$^{Y220C}$ into WT HSPCs (CD45.2$^+$) using retrovirus-mediated transduction, and then transplanted transduced cells (GFP$^+$) together with competitor bone marrow (BM) cells (CD45.1$^+$) into lethally irradiated recipient mice. We observed increased number of GFP$^+$ cells in peripheral blood (PB) of recipient mice repopulated with HSPCs expressing mutant p53 compared to that of control viruses transduced cells at 16 weeks following transplantation (Fig. 1f). Increases in total GFP$^+$ cells in PB at 16 weeks from mice with mutant p53 proteins were highly suggestive of enhanced HSPC repopulating potential.

p53 is important for HSC survival following genotoxic stress and p53 null HSCs are less sensitive to irradiation as manifested by decreased apoptosis[20]. We found that ectopic expression of p53$^{R248W}$, p53$^{R273H}$, or p53$^{Y220C}$ in WT HSPCs resulted in decreased apoptosis following 2 Gy irradiation (Fig. 1g), suggesting that HSPCs expressing these mutant p53 proteins are not sensitive to radiation-induced stress.

### *TP53* mutations in CHIP confer competitive advantage to HSPCs.

Given that codon 248 of the p53 protein (p53$^{R248}$) is most frequently mutated in CHIP, MDS, and AML[4,6,10–12,18,19], we focused our investigation on p53$^{R248W}$ in hematopoiesis. Since overexpression of mutant p53 from an MSCV promoter may not reflect accurate function when expressed at physiological levels, we utilized the *p53$^{R248W}$* knock-in mice, where p53$^{R248W}$ was introduced into the humanized *p53* knock-in (*HUPKI*) allele in mice, expressing human p53 mutant protein from the endogenous murine *Trp53* promoter[31]. The *HUPKI* allele encodes a human/mouse chimeric protein consisting primarily of human p53 sequence (amino acids 33–332) flanked by the conserved extreme amino and carboxyl-termini of mouse p53[32]. *HUPKI* mice were described as *p53$^{+/+}$* mice in the text.

Since the majority of *TP53* mutations in CHIP are mono-allele missense mutations (Fig. 1b)[4–6,10–12], we utilized heterozygous mutant mice (*p53$^{R248W/+}$*) to investigate the biological impact of mutant p53 on primitive HSPC populations. As nonsense, frameshift, and splice site mutations result in *TP53* deletions[14], we also included *p53$^{+/−}$* and *p53$^{−/−}$* mice in the experiments. We first analyzed the BM of *p53$^{+/+}$*, *p53$^{+/−}$*, *p53$^{−/−}$*, and *p53$^{R248W/+}$* mice and found that BM cellularity is comparable among these mice (Supplementary Fig. 1a). We observed increased number of

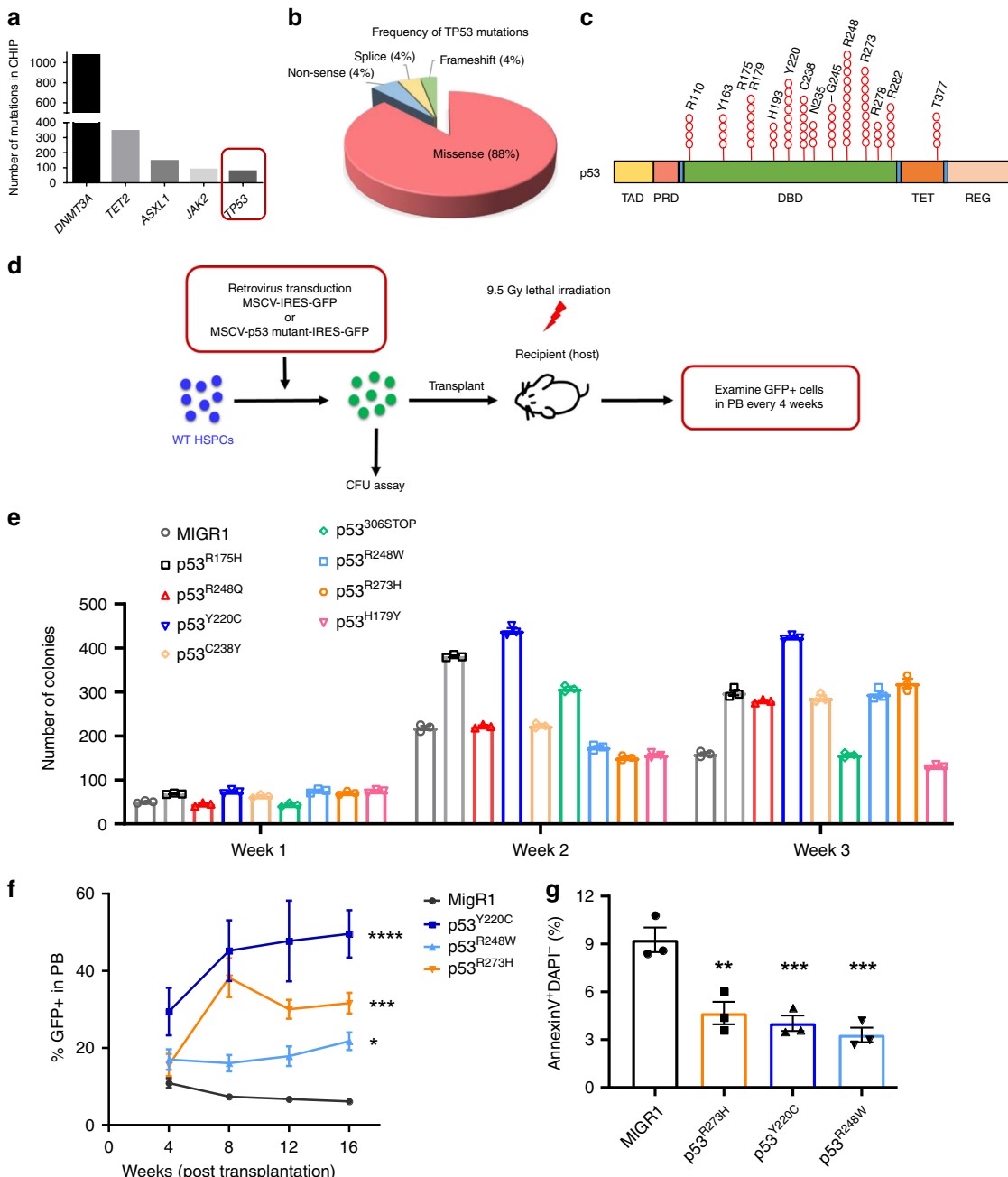

**Fig. 1 TP53 mutations identified in CHIP enhance HSPC repopulating potential. a** Tumor suppressor gene *TP53* ranks in the top five among genes that were mutated in clonal hematopoiesis with indeterminate potential (CHIP). **b** Pie chart representing different types of *TP53* mutations identified in CHIP. **c** *TP53* mutations in CHIP are enriched in the DNA-binding domain (DBD) of the p53 protein. TAD, transactivation domain; PRD, proline-rich domain; DBD, DNA-binding domain; TET, tetramerization domain; and REG, carboxy-terminal regulatory domain. **d** Several hot-spot *TP53* mutations identified in CHIP were introduced into wild-type hematopoietic stem and progenitor cells (HSPCs) using retrovirus-mediated transduction. In vitro and in vivo stem and progenitor cell assays were then performed using sorted GFP (green fluorescent protein)-postive cells. **e** Serial replating assays of HSPCs expressing different mutant p53 proteins. The methylcellulose cultures were serially replated, weekly, for 3 weeks; $n = 3$ independent experiments performed in duplicate. **f** Percentage of GFP$^+$ cells in the peripheral blood (PB) of recipient mice following competitive transplantation; $n = 3–5$ mice per group. **g** HSPCs expressing mutant p53 proteins were assessed for apoptosis at 24 h after radiation (2 Gy); $n = 3$ independent experiments. Data are represented as mean ± SEM. *P*-values were calculated using two-way ANOVA (analysis of variance) with Dunnett's multiple comparisons test in **e** and **f**, one-way ANOVA with Dunnett's multiple comparisons test in **g**; *$P < 0.05$, **$P < 0.01$, ***$P < 0.001$, ****$P < 0.0001$. Source data are provided as a Source Data file.

LSKs and LT-HSCs in the BM of $p53^{-/-}$ mice as reported[20]; however, p53$^{R248W}$ affects neither the frequency nor the number of HSPCs in the BM (Fig. 2a, b, and Supplementary Fig. 1b, c). While loss of p53 decreases the quiescence of LSKs and LT-HSCs[20], mutant p53 does not affect HSPC quiescence (Fig. 2c, d). In addition, we observed similar number of apoptotic HSPCs in

$p53^{+/+}$, $p53^{+/-}$, $p53^{-/-}$, and $p53^{R248W/+}$ mice in steady state (Fig. 2e). Thus, mutant p53 does not affect the frequency, quiescence, or the survival of HSPCs when expressed at physiological levels.

Hematopoietic transplantation is a cellular stressor that has been shown to promote the expansion of mutant HSPCs[12,33,34].

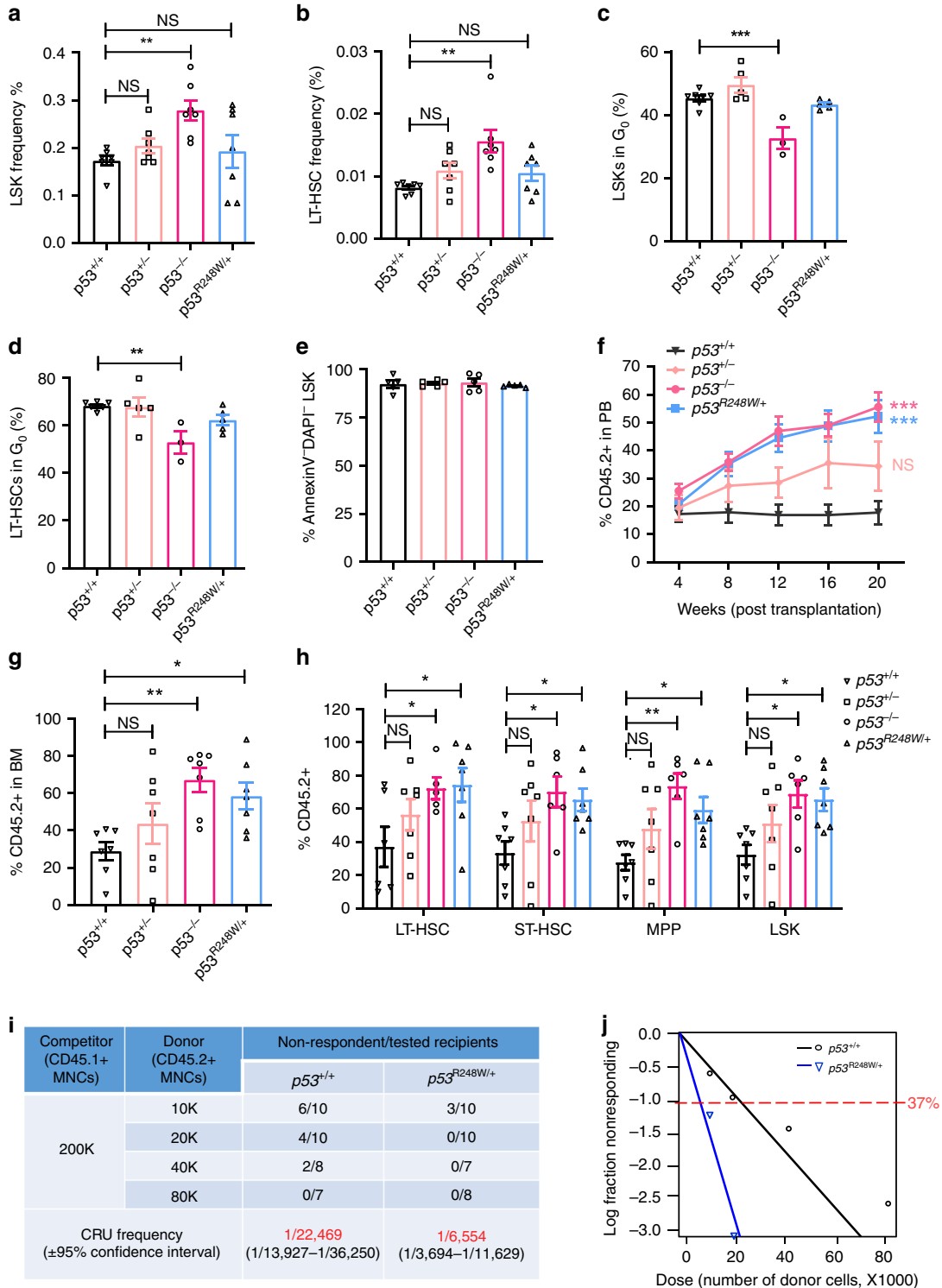

We recently reported that p53[R248W] enhances the repopulating potential of BM cells[22]. To further establish that the enhanced repopulating potential is HSC intrinsic, we purified HSCs (CD48−CD150+LSKs, CD45.2+) from p53[+/+], p53[+/−], p53[−/−], and p53[R248W/+] mice and performed HSC transplantation assays. Both p53[−/−] and p53[R248W/+] HSCs exhibited a substantially higher contribution to PB production compared to p53[+/+] and p53[+/−] HSCs at 20 weeks following primary transplantation (Fig. 2f). In addition, the percentage of donor-derived hemato-poietic cells and HSPCs in the BM of recipient mice repopulated

with p53[−/−] and p53[R248W/+] HSCs was significantly higher than that of the p53[+/+] HSCs (Fig. 2g, h). Mutant p53 did not affect myeloid and lymphoid differentiation in PB and the BM of recipient mice following HSC transplantation (Supplementary Fig. 1d, e).

To determine the impact of mutant p53 on HSC self-renewal, we performed secondary BM transplantation assays. We found that both p53[−/−] and p53[R248W/+] HSCs continue to show enhanced engraftment compared to p53[+/+] and p53[+/−] HSCs at 20 weeks following secondary transplantation. Interestingly,

**Fig. 2 $p53^{R248W/+}$ confers a competitive advantage to HSPCs. a** The frequency of Lin⁻Sca1⁺Kit⁺ cells (LSKs) in the bone marrow (BM) of $p53^{+/+}$, $p53^{+/-}$, $p53^{-/-}$, and $p53^{R248W/+}$ mice; $n = 7$ mice per genotype. **b** The frequency of long-term hematopoietic stem cells (LT-HSCs) in the BM of $p53^{+/+}$, $p53^{+/-}$, $p53^{-/-}$, and $p53^{R248W/+}$ mice; $n = 7$ mice per genotype. **c** The quiescence of LSKs was determined by Ki67 and DAPI (4′,6-diamidino-2-phenylindole) staining followed by flow cytometry analysis; $n = 3–7$ mice per genotype. **d** The quiescence of LT-HSCs was determined by Ki67 and DAPI staining and flow cytometry analysis; $n = 3–7$ mice per genotype. **e** The apoptosis of LSKs was determined by Annexin V and DAPI staining and flow cytometry analysis; $n = 5$ mice per genotype. **f** Percentage of donor-derived cells in PB of recipient mice at 20 weeks following HSC transplantation; $n = 7$ mice per group. **g** Percentage of donor-derived cells in the BM of recipient mice at 20 weeks following HSC transplantation; $n = 7$ mice per group. **h** Percentage of donor-derived LT-HSCs, short-term hematopoietic stem cells (ST-HSCs), multi-potent progenitors (MPPs), and LSKs in the BM of recipient mice repopulated with $p53^{+/+}$, $p53^{+/-}$, $p53^{-/-}$, or $p53^{R248W/+}$ HSCs; $n = 7$ mice per genotype. **i** Measuring the number of functional HSCs in the BM of $p53^{+/+}$ and $p53^{R248W/+}$ mice utilizing limiting dilution transplantation assays. Recipients with <2% donor-derived cells in the peripheral blood were defined as non-respondent; $n = 7–10$ mice per group. $P = 0.00114$. **j** Poisson statistical analysis of data from Fig. 2i using L-Calc software. Shapes represent the percentage of negative mice for each dose of cells. Solid lines indicate the best-fit linear model for each dataset. Data are represented as mean ± SEM. $P$-values were calculated using one-way ANOVA with Dunnett's multiple comparisons test in **a**, **b**, **c**, **d**, **e**, **g**, and **h**, two-way ANOVA with Dunnett's multiple comparisons test in **f**, and $\chi^2$ test in **i** and **j**; *$P < 0.05$, **$P < 0.01$, ***$P < 0.001$. Source data are provided as a Source Data file.

---

$p53^{+/-}$ HSCs show increased repopulating potential compared to $p53^{+/+}$ HSCs in secondary transplantation assays (Supplementary Fig. 1f). We observed increased number of donor-derived hematopoietic cells in the BM of secondary recipients repopulated with $p53^{+/-}$, $p53^{-/-}$, and $p53^{R248W/+}$ cells compared to $p53^{+/+}$ cells (Supplementary Fig. 1g). However, neither *TP53* mutation nor p53-deficiency alters terminal differentiation of HSCs (Supplementary Fig. 1h).

To enumerate the numbers of functional HSCs in the BM of $p53^{R248W/+}$ mice, we performed competitive BM transplantation experiments with limiting-dilution of donor cells. The frequency of competitive repopulation units (CRU) in the BM of $p53^{R248W/+}$ mice is three- to four-fold higher than that of the $p53^{+/+}$ mice (Fig. 2i, j). Enhanced repopulating potential of $p53^{R248W/+}$ BM cells could be due to changes in homing capacities of donor cells. We performed homing assays but did not detect difference in the frequency of donor-derived cells in the BM of recipient mice repopulated with $p53^{R248W/+}$ BM cells compared to $p53^{+/+}$ BM cells (Supplementary Fig. 1i, j). Taken together, we demonstrate that mutant p53 identified in CHIP confers a competitive advantage to HSPCs following transplantation.

**TP53 mutations promote HSPC survival following radiation.** Therapy-related CHIP in patients with non-hematologic cancers is common and associated with adverse clinical outcomes[10,12]. Cytotoxic therapy results in the expansion of clones carrying *TP53* mutations[10,12,35]. Indeed, we found that chemotherapy treatment expands HSPCs expressing mutant p53[22]. Given that HSPCs expressing mutant p53 are not sensitive to radiation (Fig. 1g), we then examined the impact of radiation on mutant HSPC expansion. We generated mixed BM chimeras containing both $p53^{R248W/+}$ (CD45.2⁺) and $p53^{+/+}$ (CD45.1⁺) cells with a 1:10 ratio of mutant to WT cells. Eight weeks following transplantation, recipient mice were treated with or without 5 Gy total body irradiation (TBI) (Fig. 3a). We found that mutant BM cells outcompeted $p53^{+/+}$ cells, and became clonally dominant following TBI (Fig. 3b). Further, TBI significantly increased frequency of mutant HSPCs in BM of recipient mice (Fig. 3c, d). Thus, we demonstrate that radiation promotes the expansion of HSPCs with mutant p53.

To determine the impact of radiation on $p53^{+/+}$ and $p53^{R248W/+}$ mice, we irradiated these mice and monitored their survival. While most $p53^{+/+}$ mice died 5 weeks following 9 Gy TBI, most $p53^{R248W/+}$ mice were still alive (Supplementary Fig. 2a). Further, $p53^{R248W/+}$ HSCs show decreased apoptosis both in vitro and in vivo following 2 Gy irradiation (Fig. 3e, f). Using phosphorylation of histone H2AX (γH2AX) as an indicator of DNA damage, we found that $p53^{+/+}$ HSCs stained positive for γH2AX, whereas

$p53^{R248W/+}$ HSCs were largely devoid of γH2AX foci (Supplementary Fig. 2b, c).

To determine the impact of radiation on $p53^{R248W/+}$ HSPC function in vivo, we treated $p53^{+/+}$ and $p53^{R248W/+}$ mice with 2 Gy TBI. Two hours following TBI, we isolated live BM cells from irradiated mice and performed competitive transplantation assays. Irradiated mutant BM cells displayed enhanced repopulating potential in primary transplantation assays compared to irradiated WT cells (Fig. 3g). We observed increased number of donor-derived HSPCs in the BM of primary recipient mice repopulated with live $p53^{R248W/+}$ BM cells (Fig. 3h and Supplementary Fig. 2d). Sixteen weeks after secondary transplantation, $p53^{R248W/+}$ cells continued to show increased repopulating ability (Fig. 3i). While mutant p53 had no effect on multi-lineage differentiation in PB of primary recipient mice (Supplementary Fig. 2e), we found decreased myeloid differentiation and increased B cell differentiation in the PB of secondary recipient mice repopulated with mutant BM cells (Fig. 3j). Thus, we demonstrate that *TP53* mutations identified in CHIP confer resistance to radiation, leading to the selective expansion of *TP53*-mutant HSPCs.

**HSC and AML signatures were enriched in p53 mutant HSPCs.** WT p53 is a transcription factor that activates the transcription of several target genes in HSCs, including *p21* and *Necdin*[20,21]. However, we found that mutant p53 does not alter the expression of *p21* and *Necdin* in HSCs (Fig. 4a). To understand how mutant p53 enhances HSPC self-renewal, we performed transcript profiling (using RNA-seq studies and quantitative real-time PCR (qRT-PCR) analysis) to compare gene expression in HSPCs isolated from $p53^{+/+}$ and $p53^{R248W/+}$ mice. We employed Gene Set Enrichment Assays (GSEA) to group potential mutant p53 target genes into specific pathways important in HSPC behavior. HSC and AML signatures were significantly enriched in p53 mutant HSPCs compared to $p53^{+/+}$ HSPCs (Supplementary Fig. 3a, b). Several pathways important for HSC maintenance, including Regulation of hematopoiesis, Hematopoietic organ development, Immune response, and Positive regulation of cytokine response, were significantly enriched in $p53^{R248W/+}$ HSPCs compared to $p53^{+/+}$ HSPCs (Supplementary Fig. 3c). Collectively, the gene expression profiling data suggest that mutant p53 modulates specific pathways associated with HSC maintenance and leukemogenesis.

**EZH2 targets were downregulated in p53 mutant HSPCs.** While we found that several hundred genes are either upregulated or downregulated in p53 mutant HSPCs compared to WT HSPCs, how mutant p53 regulates gene expression in HSPCs is

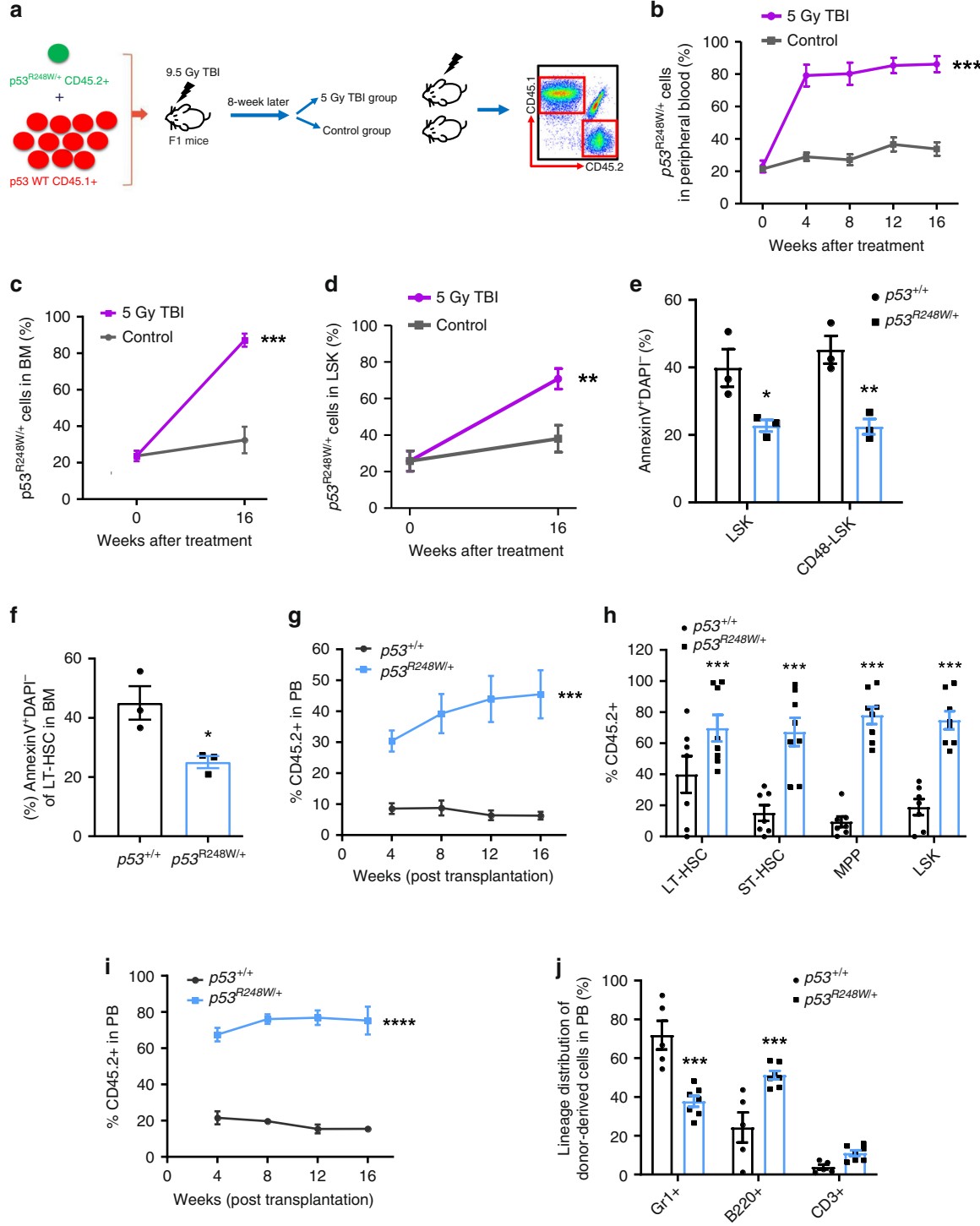

unknown. Recent studies revealed that some mutant p53 proteins increase the expression of epigenetic factors, including *MLL1*, *MLL2*, and *MOZ (KAT6A)*, in human cancer cells[36]. MLL1 and MLL2 are key components of the MLL complexes that confer histone H3K4 trimethylation (H3K4me3), which is an active histone mark important for gene expression[37,38]. MOZ is a histone acetyltransferase and mediates histone H3K9 acetylation (H3K9ac)[39,40]. However, we found that the expression of *MLL1*, *MLL2*, and *MOZ* is comparable in p53 WT and mutant HSPCs (Supplementary Fig. 3d), suggesting mutant p53 may utilize other mechanisms to modulate gene expression in hematopoietic cells.

Interestingly, RNA-seq assays revealed that genes that only marked by H3K27me3 were negatively enriched with significance in mutant HSPCs compared to that of the WT HSPCs (Fig. 4b, left panel). EZH2 target gene signature (without EZH1 compensation) was also negatively enriched with significance in mutant HSPCs (Fig. 4b, right panel). However, loss of p53 in HSPCs did not significantly change EZH2 target gene signature (Supplemental Fig. 3e). Western blot analysis showed increased levels of H3K27me3 in p53 mutant HSPCs compared to WT HSPCs (Fig. 4c). Further, $p53^{R248W/+}$ HSCs displayed higher levels of H3K27me3 compared to $p53^{+/+}$ HSCs quantified by flow cytometry analysis (Fig. 4d).

**Fig. 3** $p53^{R248W/+}$ **confers a survival advantage to HSPCs after radiation. a** BM chimeras were generated by transplanting a 1:10 ratio of $p53^{R248W/+}$ cells (CD45.2$^+$) to $p53^{+/+}$ cells (CD45.1$^+$) into irradiated recipient mice (CD45.1$^+$CD45.2$^+$). After hematopoietic reconstitution (8 weeks), mice were treated with or without 5 gray (Gy) total body irradiation (TBI). **b** Percentage of $p53^{R248W/+}$ (CD45.2$^+$) cells in PB of recipient mice following TBI treatment; $n = 7$ mice per group. **c** Percentage of $p53^{R248W/+}$ cells (CD45.2$^+$) in the BM of recipient mice at 16 weeks following TBI treatment; $n = 7$ mice per group. **d** Percentage of $p53^{R248W/+}$ LSK cells (CD45.2$^+$) in the BM of recipient mice at 16 weeks following TBI treatment; $n = 7$ mice per group. **e** Hematopoietic stem and progenitor cells from $p53^{+/+}$ and $p53^{R248W/+}$ mice were assessed for apoptosis 2 h after 2 Gy TBI; $n = 3$ mice per group. **f** HSCs purified from the BM of $p53^{+/+}$ and $p53^{R248W/+}$ mice were treated with 2 Gy TBI and then assessed for apoptosis; $n = 3$ mice per group. **g** Competitive transplantation assays using BM cells isolated from $p53^{+/+}$ and $p53^{R248W/+}$ mice treated with 2 Gy TBI. Two hours following TBI, we isolated BM cells from irradiated mice and transplanted 500,000 live BM cells together with equal number of competitor BM cells into lethally irradiated recipient mice. The percentage of donor-derived cells in the PB of recipient mice; $n = 7$–8 mice per group. **h** Percentage of donor-derived LT-HSCs, ST-HSCs, MPPs, and LSK cells in the PB of the primary recipient mice 16 weeks following transplantation; $n = 7$–8 mice per group. **i** Contribution of $p53^{+/+}$ and $p53^{R248W/+}$ BM cells to recipient mouse PB in secondary transplantation assays; $n = 5$–7 mice per group. **j** Lineage contribution of donor-derived cells in the PB of secondary recipient mice 16 weeks following transplantation; $n = 5$–7 mice per group. Data are represented as mean ± SEM. $P$-values were calculated using two-way ANOVA with Bonferroni's multiple comparisons test in **b**, **e**, **g**, **i**, and **j**, unpaired $t$-test with Welch's correction in **c**, **d**, **f**, and **h**; $^*P < 0.05$, $^{**}P < 0.01$, $^{***}P < 0.001$, $^{****}P < 0.0001$. Source data are provided as a Source Data file.

To further understand how mutant p53 modulates gene expression in hematopoietic cells, we performed H3K27me3 ChIP-seq assays in HSPCs from $p53^{+/+}$ and $p53^{R248W/+}$ mice. As expected, H3K27me3 is enriched at the transcription start site (TSS). Large regions of H3K27me3 enrichments are also found covering entire gene regions as well as intragenic regions. We found that p53 mutant HSPCs exhibited significantly higher levels of H3K27me3 at TSS compared to that of the WT HSPCs (Fig. 4e). The heat map of H3K27me3 ChIP-seq also revealed that many genes show increased H3K27me3 enrichment in mutant HSPCs compared to that of the WT HSPCs (Fig. 4f). The majority of H3K27me3 peaks (2582 out of 2669) in WT HSPCs are overlapped with H3K27me3 peaks in p53 mutant HSPCs (Supplementary Fig. 4a). By using the same enrichment threshold, we obtained 1232 additional peaks in mutant HSPCs. These peaks are likely targeted by H3K27me3 in WT cells but failed to reach the threshold. Indeed, these peaks show a similar pattern of fold enrichment between mutant and WT HSPCs (Supplementary Fig. 4b). We also observed increased H3K27me3 enrichment in other genes in p53 mutant HSPCs (Supplementary Fig. 4c, d).

Increased levels of H3K27me3 were found in genes regulating HSC self-renewal and differentiation, including *Cebpα* and *Gadd45g*[41–43], in $p53^{R248W/+}$ HSPCs compared to $p53^{+/+}$ HSPCs (Fig. 4f, g). The transcription factor C/EBP alpha is required for granulopoiesis and frequently disrupted in human AML. Loss of *Cebpα* enhances HSC repopulating capability and self-renewal[41,42]. Tumor suppressor GADD45G induces HSC differentiation following cytokine stimulation, whereas loss of GADD45G enhances the self-renewal potential of HSCs[43]. We confirmed that there were increased levels of H3K27me3 at both *Cebpα* and *Gadd45g* genes by ChIP experiments (Fig. 4h). Consistently, both *Cebpα* and *Gadd45g* were significantly down-regulated in $p53^{R248W/+}$ HSCs compared to $p53^{+/+}$ HSPCs (Fig. 4i, j).

Stimulation of WT HSPCs with thrombopoietin (TPO) dramatically increased *Gadd45g* expression; however, TPO treatment only modestly increased *Gadd45g* expression in p53 mutant HSPCs (Supplementary Fig. 5a), suggesting that mutant p53 may repress *Gadd45g* expression upon cytokine stimulation. To determine the impact of Gadd45g on HSPC function in vitro, we introduced *Gadd45g* into BM cells from $p53^{+/+}$ and $p53^{R248W/+}$ mice using retroviruses and performed colony formation as well as transplantation assays. We found that ectopic *Gadd45g* expression decreases the colony formation of p53 mutant BM cells (Supplementary Fig. 5b). Further, ectopic *Gadd45g* expression decreased the engraftment of p53 mutant BM cells in vivo (Supplementary Fig. 5c). Given that loss of Gadd45g increases HSC self-renewal[43], it is possible that inactivation of Gadd45g is responsible for increased self-

renewal and colony formation seen in p53 mutant HSPCs. We also found that ectopic *Cebpα* expression decreases the colony formation of p53 mutant BM cells (Supplementary Fig. 5d). These data suggest that mutant p53 may repress gene expression in HSPCs through increasing the levels of H3K27me3.

**Mutant p53 enhances the association of EZH2 with the chromatin.** The PRC2 complex consists of EZH2/EZH1, EED, and SUZ12[25]. While the levels of EZH2 was modestly increased in mutant HSPCs, the expression of other PRC2 core components was comparable between p53 WT and mutant HSPCs (Supplementary Fig. 6a). As the protein levels of PRC2 complex in mouse HSPCs were very low, we determined the impact of mutant p53 on the expression of PRC2 complex in murine hematopoietic progenitor 32D cells. We found that ectopic expression of mutant p53, but not WT p53, increased levels of H3K27me3 in 32D cells (Fig. 5a). However, ectopic expression of neither WT nor mutant p53 affected the protein levels of PRC2 core components in 32D cells (Supplementary Fig. 6b). Thus, the increased H3K27me3 in p53 mutant HSPCs may not be due to increased expression of catalytic components of PRC2, including EZH2 and EZH1, or other components of the PRC2 complex.

We then tested whether mutant p53 interacts with EZH2. We performed co-immunoprecipitation assays and found that several mutant p53 proteins, including p53$^{R248W}$, p53$^{R273H}$, and p53$^{R175H}$, displayed enhanced association with EZH2 compared to WT p53 (Fig. 5b). The recruitment and displacement of the PRC2 complex on chromatin are a dynamic process and tightly regulated to activate or repress transcription[23–26]. Genome-wide H3K27me3 ChIP-seq assays revealed that the majority of H3K27me3 peaks in p53 mutant HSPCs are overlapped with that of the WT HSPCs (Fig. 4e, f, Supplementary Fig. 4a), suggesting that mutant p53 may enhance the association of EZH2 with the chromatin, thereby increasing the levels of H3K27me3. To test this, we examine the co-localization of p53 and Ezh2 in p53 WT and mutant HSPCs utilizing the ImageStream flow cytometry analysis. The median fluorescent intensity (MFI) of p53 and Ezh2 in the nucleus was comparable between $p53^{+/+}$ and $p53^{R248W/+}$ HSPCs (Supplementary Fig. 6c, d). However, we found that mutant p53, but not WT p53, show increased co-localization with Ezh2 in the nucleus (Fig. 5c, d).

To determine whether mutant p53 enhances the association of Ezh2 with the chromatin, we separated proteins in p53 WT and mutant HSPCs into cytosol, nuclear cytosol, and chromatin bound fractions. While EZH2 was present in the cytosol of both p53 WT and mutant HSPCs, we observed increased levels of Ezh2 in the chromatin bound faction of p53 mutant HSPCs compared to that of the p53 WT HSPCs (Fig. 5e). We then performed p53 and EZH2 ChIP assays in $p53^{+/+}$ and $p53^{R248W/+}$ HSPCs

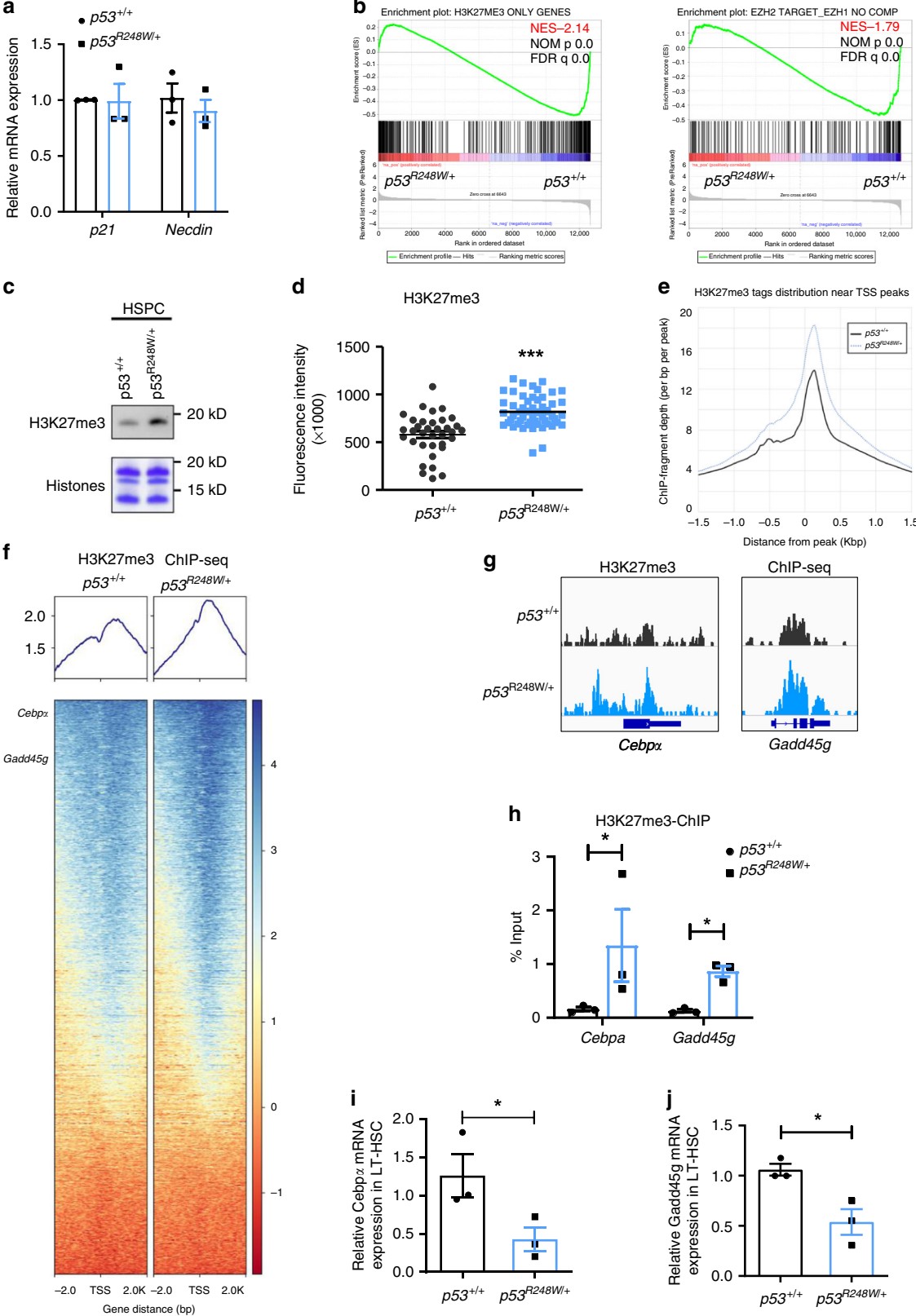

(Lin⁻Kit⁺ cells) and found that both mutant p53 and EZH2 show increased association with *Cebpα* (Supplementary Fig. 6e, f). Thus, we demonstrate that mutant p53 interacts with EZH2 and enhances its association with the chromatin, thereby increasing the levels of H3K27me3 in HSPCs.

**Inhibiting EZH2 decreases p53 mutant HSPC expansion.** Hematopoietic-specific deletion of *Ezh2* impairs HSC self-renewal and terminal differentiation[28]. To determine the functional impacts of mutant p53 and Ezh2 interaction on hematopoiesis, we generated $p53^{R248W/+}Ezh2^{f/f}-Mx1Cre^+$ mice[28]. While Ezh2-

**Fig. 4 EZH2 targets were significantly downregulated in p53 mutant HSPCs. a** Quantitative reverse transcription polymerase chain reaction (RT-PCR) analysis of mRNA levels of p53 target genes, including *p21* and *Necdin*, in HSCs; $n = 3$ biological replicates. **b** Gene Set Enrichment Assays (GSEA) analysis shows that EZH2 targets were significantly downregulated in p53 mutant HSPC compared to $p53^{+/+}$ HSPCs. **c** p53 mutant HSPCs display increased levels of H3K27me3 (trimethylation at lysine 27 of histone H3) determined by immuno-blot analysis. **d** Lineage depleted HSPCs were stained with SLAM (signaling lymphocyte activation molecule) surface markers (CD48 and CD150) before fixation. Median fluorescence intensity of H3K27me3 in $p53^{+/+}$ and $p53^{R248W/+}$ HSCs (Lin⁻Sca1⁺Kit⁺CD48⁻CD150⁺ cells) was detected by ImageStream flow cytometry analysis. $p53^{+/+}$ $n = 35$ cells, $p53^{R248W/+}$ $n = 52$ cells. **e** H3K27me3 ChIP-seq (chromatin immunoprecipitation sequencing) tag density in $p53^{+/+}$ and $p53^{R248W/+}$ HSPCs, centered on TSS (transcription start site). **f** Heat map shows genes in HSPCs marked by H3K27me3. **g** Genome browser views of H3K27me3 ChIP-seq profiles of *Cebpα* (CCAAT/enhancer-binding protein alpha) and *Gadd45g* (growth arrest and DNA-damage-inducible 45 gamma). **h** H3K27me3 enrichment on *Cebpα* and *Gadd45g* genes in $p53^{+/+}$ and $p53^{R248W/+}$ HSPCs were examined by H3K27me3-ChIP assays; $n = 3$ independent experiments. **i** Quantitative RT-PCR analysis of mRNA levels of *Cebpα* in $p53^{+/+}$ and $p53^{R248W/+}$ LT-HSCs; $n = 3$ biological replicates. **j** Quantitative RT-PCR analysis of mRNA levels of *Gadd45g* in $p53^{+/+}$ and $p53^{R248W/+}$ LT-HSCs; $n = 3$ biological replicates. Data are represented as mean ± SEM. *P*-values were calculated using unpaired *t* test with Welch's correction in **a** and **d**, paired *t*-test in **h**, **i**, and **j**, and GSEA software in **b**; *$P < 0.05$, ***$P < 0.001$. Source data are provided as a Source Data file.

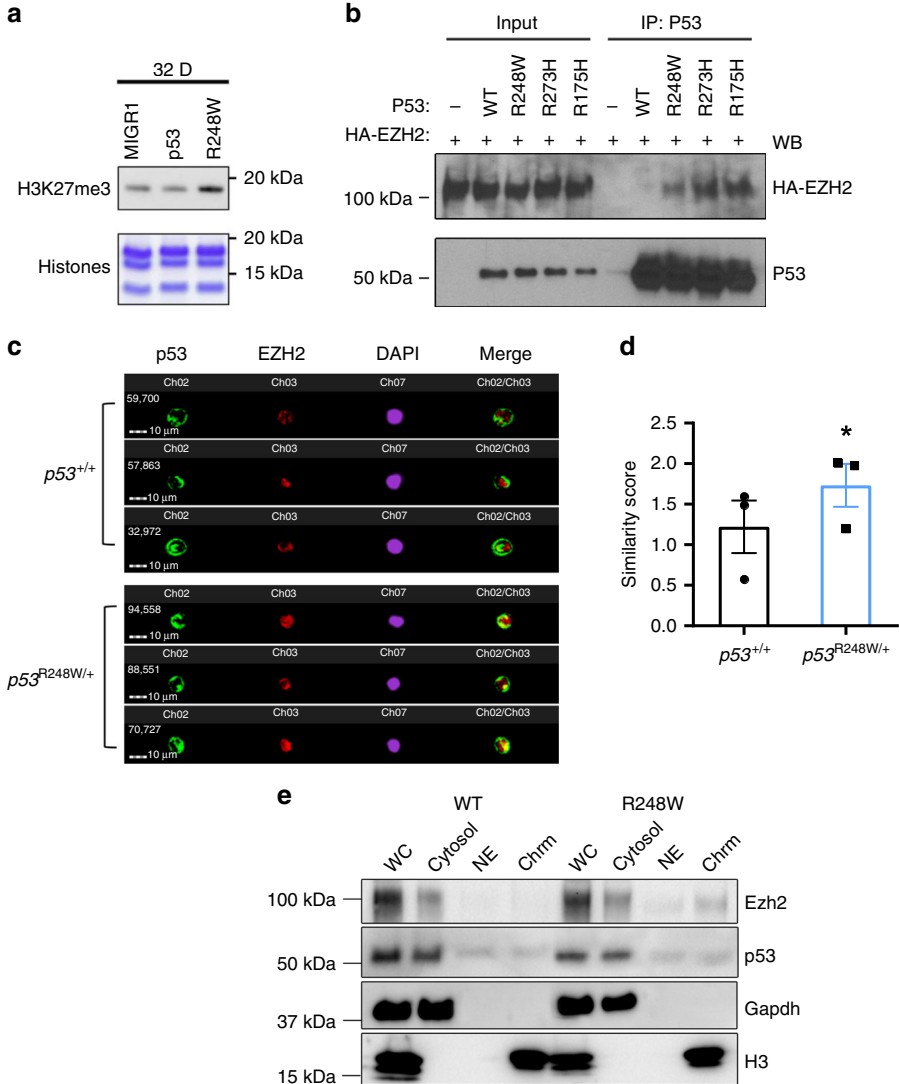

**Fig. 5 Mutant p53 enhances the association of EZH2 with the chromatin in HSPCs. a** 32D cells expressing mutant p53, but not wild-type (WT) p53, displayed increased levels of H3K27me3 as determined by immuno-blot analysis. **b** Several mutant p53 proteins, but not wild-type p53, show enhanced association with EZH2 as assayed by co-IP (co-immunoprecipitation) experiments. **c** Mutant p53 and EZH2 localization in HSPCs (Lin⁻Sca1⁺Kit⁺CD150⁺) as determined by ImageStream flow cytometry analysis. **d** Quantification of p53 and Ezh2 co-localization in the nucleus of HSPCs (Lin⁻Sca1⁺Kit⁺CD150⁺). A similarity feature determined the amount of overlay between p53 and Ezh2 within the DAPI mask. The higher the similarity score is, the more co-localized staining is within the nucleus; $n = 3$ biological replicates. **e** Cellular fractionation shows increased EZH2 association with the chromatin fraction in p53 mutant HSPCs. The absence of Gapdh (glyceraldehyde 3-phosphate dehydrogenase) and exclusive distribution of histone H3 in the chromatin fraction indicates no cross contamination between different cellular compartments. WC whole cell extract, Cyto cytosol, NE nuclear cytosol, Chrm chromatin. Data are represented as mean ± SEM. *P*-values were calculated using paired *t*-test in **d**; *$P < 0.05$. Source data are provided as a Source Data file.

deficiency did not affect *Cebpα* and *Gadd45g* expression, Ezh2 deficiency brought the expression of *Cebpα* and *Gadd45g* back to normal in the mutant p53 background (Fig. 6a and Supplementary Fig. 7a).

To determine the impact of genetic inhibition of Ezh2 on p53 mutant HSPC function in vitro, we performed serial replating assays using BM cells from $p53^{+/+}$, $Ezh2^{f/+}$-*Mx1Cre*+, $p53^{R248W/+}$, and $p53^{R248W/+}Ezh2^{f/+}$-*Mx1Cre*+ mice following polyinosinic: polycytidylic acid (pI:pC) treatment. We found that Ezh2-deficiency brings the replating potential of p53 mutant BM cells back to the WT cell level (Fig. 6b). To determine the impact of Ezh2 deficiency on HSPC function in vivo, we performed competitive BM transplantations. We treated recipient mice with pI:pC at 8 weeks after transplantation to delete Ezh2 and then examined donor cell engraftment every 4 weeks for 20 weeks. $p53^{R248W/+}$ BM cells exhibited a substantially higher contribution to PB production compared to $p53^{+/+}$ cells at 20 weeks following pI:pC treatment, whereas loss of Ezh2 decreased the engraftment of mutant BM cells to the WT cell level (Fig. 6c). While both the frequency and the absolute number of donor-derived HSPCs in the BM of recipient mice repopulated with $p53^{R248W/+}$ BM cells was significantly higher than that of the WT cells, the frequency and the number of donor-derived HSPCs in the BM of recipient mice repopulated with $p53^{+/+}$ and $p53^{R248W/+}Ezh2^{-/-}$ BM cells were comparable (Fig. 6d, e). In addition, we observed decreased number of donor-derived CMPs and MEPs in the BM of recipient mice repopulated with $p53^{R248W/+}Ezh2^{-/-}$ BM cells compared to that of the $p53^{R248W/+}$ BM cells (Supplementary Fig. 7b, c), whereas the number of donor-derived GMPs was comparable in the BM of recipient mice repopulated with $p53^{R248W/+}$ cells and $p53^{R248W/+}Ezh2^{-/-}$ cells (Supplementary Fig. 7d). To determine the impact of Ezh2-deficiency on mutant HSC self-renewal, we performed secondary BM transplantation assays and found that EZH2-deficiency decreases the repopulating potential of $p53^{R248W/+}$ HSCs following secondary transplantation assays (Fig. 6f).

To determine the effect of pharmacological inhibition of Ezh2 activity on mutant p53 HSPCs, we treated $p53^{+/+}$ and $p53^{R248W/+}$ BM cells with DMSO or EZH2 specific inhibitor EPZ011989[44] and performed serial replating assays. While EZH2 inhibitor had no effect on the colony formation of WT BM cells, EZH2 inhibitor treatment decreased the replating potential of $p53^{R248W/+}$ BM cells to the WT level (Fig. 6g). Thus, we demonstrate that EZH2 is important for mutant p53 HPSC functions both in vitro and in vivo.

## Discussion

WT p53 is a transcription factor that activates the transcription of target genes to mediate DNA damage repair, growth arrest, or apoptosis[13,45]. Most TP53 mutations observed in human cancers abrogate or attenuate the binding of p53 to its consensus DNA sequence (p53 responsive element) and impede transcriptional activation of p53 target genes[14]. However, we found that mutant p53 does not alter the expression of p53 target genes, including p21 and Necdin, in HSCs (Fig. 4a). Genome-wide transcriptome assays revealed that HSC and AML signatures are enriched in p53 mutant HSPCs, which is different from gene expression signatures regulated by the WT p53 protein[20,45]. Thus, our findings provide experimental evidence that TP53 mutations identified in CHIP regulate gene expression in a distinct manner compared to WT p53.

Some mutant p53 proteins have been shown to promote cancer development through modulating gene transcription[14]. Dysregulated epigenetic control has been implicated in HSC aging and the pathogenesis of hematological malignancies[16,46–48]. RNA-seq assays revealed that Ezh2 target genes are significantly downregulated in p53R248W/+ HSPCs compared to p53+/+ HSPCs. We observed increased levels of H3K27me3 in p53 mutant HSPCs. Further, H3K27me3 ChIP-seq assays revealed that p53 mutant HSPCs exhibit significantly high levels of H3K27me3. Genes important for HSC self-renewal and differentiation, including Cebpα and Gadd45g[41–43], were occupied with increased levels of H3K27me3 in p53 mutant HSPCs. Thus, mutant p53 may enhance HSPC self-renewal through increasing the levels of H3K27me3 in genes involved in HSC self-renewal and differentiation.

Then, how does mutant p53 enhances H3K27me3 in HSPCs? One possible mechanism is that mutant p53 upregulates the expression of the core PCR2 components, thereby increasing PRC2 activity. However, the expression of the core PRC2 components was comparable between p53 WT and mutant HSPCs. Mutant p53 is incapable of binding to its normal binding sites and has been shown to be targeted by interactions with other transcription factors, including ETS family and SREBP[14,36]. We discovered that several mutant p53 proteins show enhanced association with EZH2 compared to WT p53. H3K27me3 ChIP-seq assays revealed that the increase in EZH2-dependent H3K27me3 is broad across genes, pointing toward an alternative EZH2/PRC2 targeting strategy or an increase in enzymatic activity but with normal targeting mechanisms. As shown in Supplementary Fig. 4a, 96.7% H3K27me3 peaks in WT cells also show H3K27me3 occupation in p53 mutant cells. Although additional peaks are enriched in p53 mutant cells, they are indeed associated with H3K27me3 in WT cells as well albeit lower enrichment. Further, we found that mutant p53 interacts with EZH2 and enhances its association with the chromatin in HSPCs. Thus, mutant p53 appears to enhance H3K27me3 occupation rather than change its genome-wide distribution in HSPCs.

While the PRC2 complex controls dimethylation and trimethylation of H3K27, the Jumonji domain containing-3 (Jmjd3, KDM6B) and ubiquitously transcribed X-chromosome tetratricopeptide repeat protein (UTX, KDM6A) have been identified as H3K27 demethylases that catalyze the demethylation of H3K27me2/3[49,50]. Decreased activity of KDM6A/UTX and JMJD3 may be an alternative mechanism leading to increased EZH2 activity. Future studies will be needed to investigate the potential impact of UTX and JMJD3 on regulating H3K27me3 in p53 mutant HSPCs.

Most of TP53 mutations in human cancer result in either partial or complete loss of tumor suppressor function[14]. Some mutant p53 proteins acquire new oncogenic properties that are independent of WT p53, known as the gain-of-function (GOF) properties[14]. Most GOF properties are believed to stem from binding of mutant p53 to cellular proteins such as transcription factors and altering their activity[14]. These (neomorphic) GOF properties can be experimentally demonstrated in the absence of a functional WT p53. Homozygous $p53^{R248W/R428W}$ and $p53^{R273H/R273H}$ mice developed novel tumors compared to $p53^{-/-}$ mice[29,31], demonstrating that some mutant p53 proteins have enhanced oncogenic potential beyond the simple loss of p53 function. WT p53 has not been shown to be associated with EZH2 activity or H3K27me3. RNA-seq assays revealed that PRC2-related gene signature was not significantly different between $p53^{+/+}$ and $p53^{-/-}$ HSPCs, which is different from what we observed in p53 mutant HSPCs, suggesting that loss of WT p53 may affect neither EZH2 activity nor H3K27me3 in HSPCs. Ectopic expression of mutant p53, but not WT p53, enhances H3K27me3 in 32D cells. Further, we found that several mutant p53 proteins show enhanced association with EZH2 compared to WT p53. While both loss of p53 ($p53^{-/-}$) and mutant p53 ($p53^{R248W/+}$) enhances HSC repopulating potential, our data

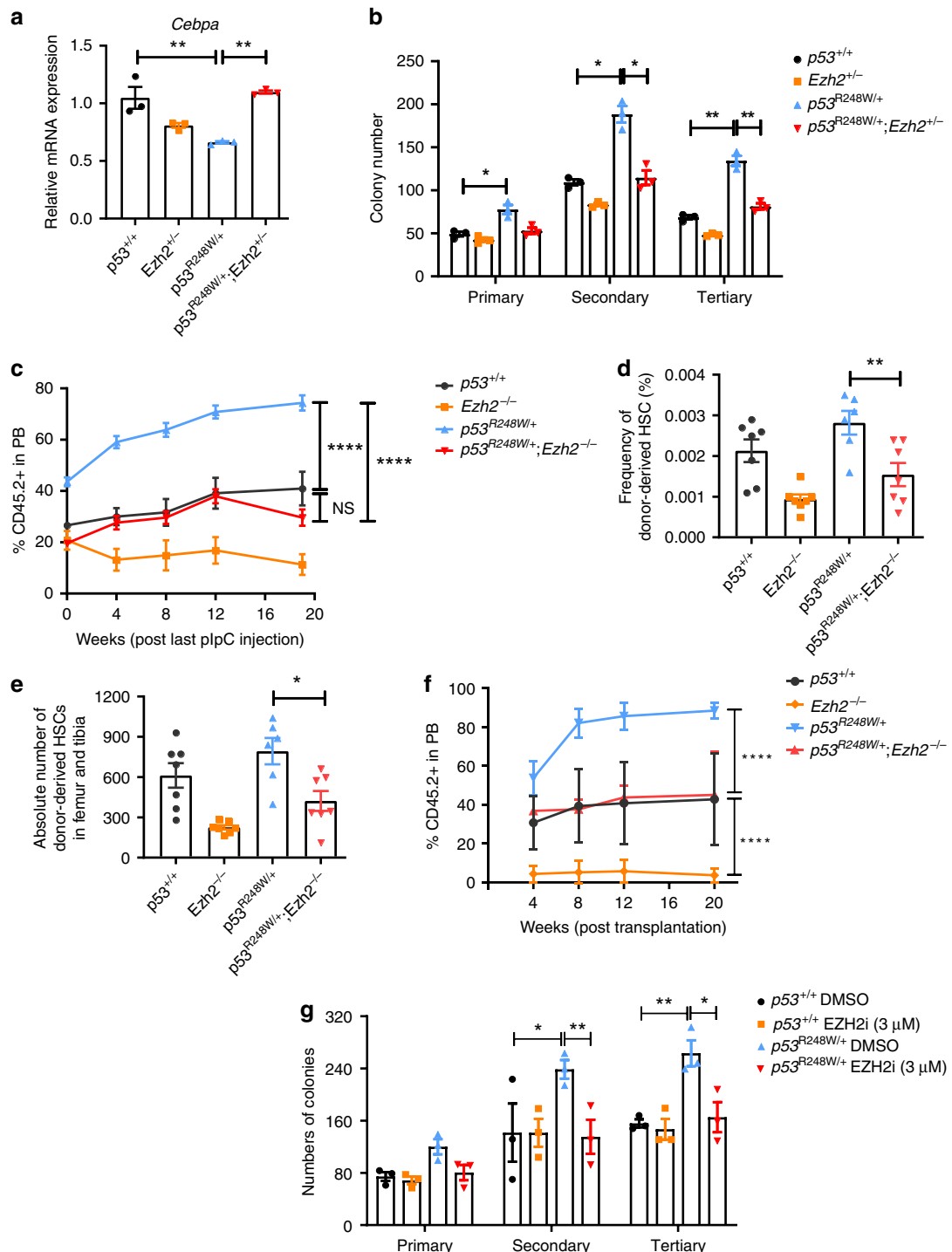

**Fig. 6 Loss of EZH2 decreases the repopulating potential of p53 mutant HSPCs. a** *Cebpα* expression in *p53*^+/+, *Ezh2*^+/−, *p53*^R248W/+, and *p53*^R248W/+ *Ezh2*^+/− HSPCs; *n* = 3 biological replicates. **b** Serial replating assays of BM cells from *p53*^+/+, *Ezh2*^+/−, *p53*^R248W/+ and *p53*^R248W/+ *Ezh2*^+/− mice; *n* = 3 independent experiments. **c** Percentage of donor-derived cells in the PB of recipient mice at 20 weeks following pI:pC (polyinosinic:polycytidylic acid) treatment; *n* = 7 mice per group. **d** Percentage of donor-derived HSCs in the BM of recipient mice at 20 weeks following pI:pC treatment; *n* = 6-7 mice per group. **e** The absolute number of donor-derived HSCs in the BM of recipient mice at 20 weeks following pI:pC treatment; *n* = 6-7 mice per group. **f** Percentage of donor-derived cells in the PB of recipient mice at 20 weeks following secondary transplantation; *n* = 7 mice per group. **g** Serial replating assays using *p53*^+/+ and *p53*^R248W/+ BM cells treated with DMSO (dimethyl sulfoxide) or EZH2 inhibitor (3 μM); *n* = 3 independent experiments. Data are represented as mean ± SEM. *P*-values were calculated using one-way ANOVA with Tukey's multiple comparisons test in **a**, **d**, and **e**, two-way ANOVA with Tukey's multiple comparison test in **b**, **c**, **f**, and **g**; *$P < 0.05$, **$P < 0.01$, ****$P < 0.0001$. Source data are provided as a Source Data file.

suggest that increased levels of H3K27me3 in p53 mutant HSPCs is likely due to the presence of the mutant allele, but not the result of losing WT p53 activity. While a dominant-negative (DN) effect has been shown to drive selection of *TP53* missense mutations in myeloid malignancies[51], GOF mutant p53 appears to play an important role in myeloid leukemia[30]. Our work suggests that both DN and GOF properties may contribute to enhanced HSC self-renewal seen in *p53*^R248W/+ mice.

Clinical studies revealed that hematopoietic clones harboring specific mutations in individuals with CHIP may expand over time[4–6]. However, how different cellular stressors affect clonal expansion is largely unknown. Recently, two different stressors, including hematopoietic transplantation and cytotoxic therapy, have been shown to expand hematopoietic clones[10,12,22,33–35]. We discovered that *TP53* mutations identified in CHIP confer a competitive advantage to HSPCs following transplantation. *TP53* mutations are associated with prior exposure to chemotherapy[10,12] and we observed that *TP53* mutations confer radiation resistance, leading to selective expansion of *TP53*-mutant HSPCs. Recently, *PPM1D* mutations were found in CHIP, especially in patients previously exposed to chemotherapy[10,12,33,34]. PPM1D is a phosphatase that negatively regulates p53 and several proteins involved in the DDR pathway[52,53]. While *PPM1D* mutations result in the expansion of *PPM1D*-mutant hematopoietic cells following chemotherapy treatment[33,34], they do not confer competitive advantage to HSPCs following BM transplantation[12,33,34]. Thus, p53 and PPM1D appear to play distinct roles in driving clonal hematopoiesis.

While we have identified a stem cell intrinsic mechanism by which mutant p53 drives clonal hematopoiesis, recent studies indicate that mutations identified in CHIP may utilize cell extrinsic mechanisms to promote clonal hematopoiesis[54,55]. We will investigate the cell extrinsic mechanisms by which mutant p53 drives CHIP in the future. Some individuals with CHIP developed AML with age[2,3]. However, the role of mutant p53 in the initiation and progression of AML is largely unknown[18,56]. We recently reported that mutant p53 synergizes with FLT3-ITD in leukemia development[57]. We will elucidate the mechanisms by which mutant p53 drives leukemia development.

In summary, we discovered that *TP53* mutations drive clonal hematopoiesis in response to distinct cellular stressors. Mechanistically, mutant p53 interacts with EZH2 and enhances its association with the chromatin, increasing the levels of H3K27me3 in genes regulating HSPC self-renewal and differentiation. EZH2 is rarely mutated in CHIP[4–6] and we found that genetic and pharmacological inhibition of EZH2 decrease the repopulating potential of p53 mutant HSPCs. Thus, our work will likely establish epigenetic regulator EZH2 as a novel therapeutic target for preventing CHIP progression and treating hematological malignancies with *TP53* mutations.

## Methods

**Mice**. The HUPKI ($p53^{+/+}$) and $p53^{R248W/+}$ mice used in our studies have been backcrossed to the C57BL6 background for 12 generations[22,31]. All young $p53^{+/+}$, $p53^{+/-}$, $p53^{-/-}$, and $p53^{R248W/+}$ and *Ezh2$^{F/F}$-Mx1-Cre$^+$* mice used in these studies are 8–12 weeks old and are tumor free. WT C57BL/6 (CD45.2$^+$), B6.SJL (CD45.1$^+$), and F1 mice (CD45.2$^+$ CD45.1$^+$) mice were obtained from an on-site core breeding colony. We have complied with all relevant ethical regulations for animal testing and research. All animal-related experiments have received ethical approval from the Indiana University Institutional Animal Care and Use Committee (IACUC). All mice were maintained in the Indiana University Animal Facility according to IACUC-approved protocols.

**Generation of retroviruses and infection of HSPCs**. Retroviral vectors were produced by transfection of Phoenix E cells with the MIGR1 control or MIGR1 full-length mutant p53 cDNA plasmids, according to standard protocols. Mouse HSPCs were infected with high-titer retroviral suspensions in the presence of Retronectin. Forty-eight hours after infection, the GFP-positive cells were sorted by FACS[20,21].

**Stem and progenitor cell assays**. Clonogenic progenitors were determined in methylcellulose medium (MethoCult GF M3234, StemCell Technologies) using $2 \times 10^4$ BM cells per well (6-well plate). Colonies were scored after 7 days of the initial culture, and all cells were collected and washed twice in PBS. Subsequently cells were cultured in the same medium. Colony scoring and replating were repeated every 7 days for at least two times[20,21].

**Flow cytometry**. Murine HSPCs were identified and evaluated by flow cytometry using a single cell suspension of bone marrow mononuclear cells (BMMCs). Hematopoietic stem and progenitors are purified based upon the expression of surface markers: LT-HSC (Lin$^-$Sca1$^+$Kit$^+$CD48$^-$CD150$^+$), ST-HSC (Lin$^-$Sca1$^+$Kit$^+$CD48$^-$CD150$^-$), MPP (Lin$^-$Sca1$^+$Kit$^+$CD48$^+$CD150$^-$), CMP (Lin$^-$Sca1$^-$Kit$^+$CD16/32w$^{low}$CD34$^{high}$), GMP (Lin$^-$Sca1$^-$Kit$^+$CD16/32w$^{high}$CD34$^{high}$), and MEP (Lin$^-$Sca1$^-$Kit$^+$CD16/32w$^{low}$CD34$^{low}$). BM cells were obtained from tibia, femur, and iliac crest (6 from each mice) by flushing cells out of the bone using a syringe and phosphate-buffered saline (PBS) with 2 mM EDTA. Red blood cells (RBCs) were lysed by RBC lysis buffer (eBioscience) prior to staining. Experiments were performed on FACS LSR IV cytometers (BD Biosciences) and analyzed by using the FlowJo software (TreeStar).

**Ki-67 staining**. BM cells were stained for cell surface markers as described above. After staining, cells were washed with 0.2% BSA in PBS, fixed and permeabilized using Cytofix/Cytoperm buffer (BD Biosciences) and then incubated with PE-conjugated-antibody against Ki-67 (BD Biosciences) for more than 30 min on ice. Cells were washed, incubated with 4′,6-diamidino-2-phenylindole (DAPI) (Sigma) and acquired using LSR IV flow cytometer machine[20,21]. Data analysis was performed using FlowJo software.

**Hematopoietic cell transplantation**. For HSC transplantation, we injected 200 CD48$^-$CD150$^+$LSK cells from $p53^{+/+}$, $p53^{+/-}$, $p53^{-/-}$, and $p53^{R248W/+}$ mice (CD45.2$^+$) plus $3 \times 10^5$ competitor BM cells (CD45.1$^+$) into lethally irradiated F1 mice (CD45.1$^+$CD45.2$^+$). The percentage of donor-derived (CD45.2$^+$) cells in PB was analyzed every 4 weeks after transplantation as described above. Twenty weeks following transplantation, we harvested BM cells from recipient mice and performed flow cytometry analysis to evaluate HSC repopulating capability. For secondary transplantation assays, $3 \times 10^6$ BM cells from mice repopulated with $p53^{+/+}$, $p53^{+/-}$, $p53^{-/-}$, and $p53^{R248W/+}$ HSCs were transplanted into lethally irradiated F1 mice.

For the competitive BM repopulation assays, we injected $5 \times 10^5$ BM cells from $p53^{+/+}$ and $p53^{R248W/+}$ mice (CD45.2$^+$) plus $5 \times 10^5$ competitor BM cells (CD45.1$^+$) into 9.5 Gy lethally irradiated F1 mice (CD45.1$^+$CD45.2$^+$). PB was obtained by tail vein bleeding every 4 weeks after transplantation, RBC lysed, and the PB mononuclear cells stained with anti-CD45.2 FITC and anti-CD45.1 PE, and analyzed by flow cytometry. Sixteen weeks following transplantation, BM cells from recipient mice were analyzed to evaluate donor chimerism in BMs. For secondary transplantation, $3 \times 10^6$ BM cells from mice reconstituted with $p53^{+/+}$ or $p53^{R248W/+}$ BM cells were injected into 9.5 Gy lethally irradiated F1 mice (CD45.1$^+$CD45.2$^+$).

**Limiting dilution assays**. Different doses (10,000, 20,000, 40,000, 80,000) of BM cells from $p53^{+/+}$ and $p53^{R248W/+}$ mice (CD45.2$^+$) together with 200,000 competitor cells (CD45.1$^+$) were transplanted into lethally irradiated (9.5 Gy) F1 recipient mice (CD45.2$^+$CD45.1$^+$). The percentage of donor-derived (CD45.2$^+$) cells were analyzed 16-weeks following transplantation as described above. HSC frequency was calculated using L-Calc software (StemCell Technologies Inc.) and ELDA software (bioinf.wehi.edu.au/software/elda/). Poisson statistics was used to calculate the *P* value.

**Homing assays**. A total of $1 \times 10^7$ $p53^{+/+}$ and $p53^{R248W/+}$ BM cells (CD45.2$^+$) were injected into lethally irradiated recipient mice (CD45.1$^+$). BM cells were harvested 18 h following injection and the frequency of donor-derived cells (CD45.2$^+$) was evaluated by flow cytometry.

**Quantitative real-time PCR**. Total RNA was extracted from cells using RNeasy Plus Micro Kit (Qiagen) and cDNA was prepared from total RNA using SuperScript IV First-Strand cDNA Synthesis Kit (Invitrogen Life Technologies) and oligo (dT) primers, following manufacturer's instructions. qRT-PCR assay was performed by using the 7500 Real Time PCR machine (Applied Biosystems) with FastStart Universal SYBR Green Master (ROX) (Roche).

**ImageStream flow cytometry analysis**. To quantify γ-H2Ax foci in HSPCs, lineage-depleted BM cells were first stained with antibodies against appropriate HSPC surface markers, then fixed and permeabilized using the Cytofix/Cytoperm Kit (BD Biosciences), as described by the manufacturer, and finally stained with an Alexa-488-conjugated anti-γ-H2Ax antibody (Cell Signaling Technology).

To quantify the intensity of H3K27me3 in HSPCs, lineage-depleted BM cells were first stained with antibodies against HSPC surface markers, then fixed and permeabilized using the Cytofix/Cytoperm Kit (BD Biosciences), as described by the manufacturer, and finally stained with an Alexa-488-conjugated anti-H3K27me3. For quantitative image analysis of p53 and Ezh2 co-localization within the nucleus, fluorescent cell images (×40) were acquired using an ImageStream flow cytometry system (Amnis; Seattle, WA, http://www.amnis.com). Between 171 and 554 Lin$^-$Sca1$^+$cKit$^+$CD150$^+$ cell images were analyzed per sample using IDEAS software (Amnis; Seattle, WA, http://www.amnis.com). In focus cells were evaluated after gating on live, single, Lin$^-$Sca1$^+$cKit$^+$CD150$^+$ cells. Utilizing DAPI staining, we were able to create a nucleus mask and instruct the program to only look at the staining of p53 and Ezh2 within the DAPI/nucleus mask. Bright detail intensity of

FITC-p53, PE-Ezh2, and DAPI staining was used to quantify mean and geo mean intensity and co-localization within the nucleus. A similarity feature determined the amount of overlay between p53 and Ezh2 within the DAPI mask. The higher the similarity score is, the more co-localized the staining within the nucleus.

**Co-IP**. H1299 cells (p53 null) were co-transfected with FLAG-HA-EZH2 and WT or mutant p53, respectively, or transfected with FLAG-HA-EZH2 alone. Nuclear extract (NE) was prepared from these cells and incubated with a polyclonal p53 antibody (FL393, Santa Cruz) prior to addition of protein G beads. After overnight incubation, beads were then washed five times and eluted with glycine (0.1 M, pH 2.0), and then neutralized by adding Tris solution (1.5 M, pH 8.8). The eluates were mixed with SDS sample buffer and analyzed by SDS-PAGE, followed by immunoblotting[58].

**Cellular fractionation**. Briefly, $p53^{+/+}$ and $p53^{R248W/+}$ HSPCs were harvested and lysed in Buffer A (10 mM Tris-HCl, pH 7.9, 1.5 mM $MgCl_2$, 10 mM KCl, 0.5 mM DTT, 0.2 mM PMSF, 1 µg/ml pepstatin A, 1 µg/ml leupeptin, 1 µg/ml aprotinin). The cell lysate was then homogenized by a dounce homogenizer for 10 strokes and centrifuged at 4 °C and 15,000×g for 10 min. The supernatant was saved as the cytosolic fraction. The pellet was washed with Buffer A and then re-suspended with Buffer C (20 mM Tris-HCl, pH 7.9, 25% glycerol, 420 mM NaCl, 1.5 mM $MgCl_2$, 0.2 mM EDTA, 0.5 mM DTT, 0.2 mM PMSF, 1 µg/ml pepstatin A, 1 µg/ml leupeptin, 1 µg/ml aprotinin) and dounced for 10 strokes. Suspension was rotated at 4 °C for 30 min and centrifuged at 4 °C and 7500×g for 5 min. The supernatant was saved as the nucleoplasmic fraction. The pellet was saved as the chromatin fraction and re-suspended with Buffer C. Cell fractions were mixed with SDS sample buffer and heated at 95 °C for 5 min[59]. Whole-cell extraction, nucleoplasmic, and chromatin extractions were sonicated for 15 s using a probe sonicator before loading to SDS-PAGE.

**RNA sequencing**. Total RNA is extracted from LSKs using RNeasy MicroPlus Kit (Qiagen). Then the mRNA is enriched with the oligo(dT) magnetic beads (for eukaryotes), and is fragmented into short fragments (about 100 bp). With random hexamer-primer, the first strand of cDNA is synthesized, and then the second strand is synthesized. The double-strand cDNA is purified with magnetic beads. The ends of the double strand cDNA are repaired, and a single nucleotide A (adenine) is added to the 3′-ends. Finally, sequencing adaptors are ligated to the fragments. The ligation products are amplified with PCR. For quality control, RNA and library preparation integrity are verified using Agilent 2100 BioAnalyzer system and ABI StepOnePlus Real-Time PCR System. RNA sequencing library was then constructed and then sequenced with Hiseq 4000.

Gene Set Enrichment Analysis (GSEA) was performed on gene sets from the Molecular Signatures Database (MSigDB, https://www.broadinstitute.org/msigdb) and additional gene sets curated from publications. Gene sets with FDR $q$-value <0.05 were considered significantly enriched.

**ChIP sequencing**. Lin⁻Kit⁺ cells were fixed with 1% formaldehyde for 15 min and quenched with 0.125 M glycine. Chromatin was isolated by the addition of lysis buffer, followed by disruption with a Dounce homogenizer. Lysates were sonicated and the DNA sheared to an average length of 300–500 bp. Genomic DNA (Input) was prepared by treating aliquots of chromatin with RNase, proteinase K and heat for de-crosslinking, followed by ethanol precipitation. Pellets were resuspended and the resulting DNA was quantified on a NanoDrop spectrophotometer. An aliquot of chromatin (10 µg, spiked-in with 200 ng of Drosophila chromatin) was precleared with protein A agarose beads (Invitrogen). Genomic DNA regions of interest were isolated using 4 µg of antibody against Histone H3K27me3 (clone: 39155, Active Motif). Antibody against H2Av (0.4 µg) was also present in the reaction to ensure efficient pull-down of the spike-in chromatin[60]. Complexes were washed, eluted from the beads with SDS buffer, and subjected to RNase and proteinase K treatment. Crosslinks were reversed by incubation overnight at 65 °C, and ChIP DNA was purified by phenol–chloroform extraction and ethanol precipitation.

Illumina sequencing libraries were prepared from the ChIP and Input DNAs by the standard consecutive enzymatic steps of end-polishing, dA-addition, and adaptor ligation. After a final PCR amplification step, the resulting DNA libraries were quantified and sequenced on Illumina's NextSeq 500 (75 nt reads, single end). Reads were aligned consecutively to the mouse genome (mm10) and to the Drosophila genome (dm3) using the BWA algorithm (default settings). Duplicate reads were removed and only uniquely mapped reads (mapping quality > = 25) were used for further analysis. The number of mouse alignments used in the analysis was adjusted according to the number of Drosophila alignments that were counted in the samples that were compared. Mouse alignments were extended in silico at their 3′-ends to a length of 200 bp, which is the average genomic fragment length in the size-selected library, and assigned to 32-nt bins along the genome. The resulting histograms (genomic "signal maps") were stored in bigWig files. H3K27me3 enriched regions were identified using the SICER algorithm with a MaxGap parameter setting of 600 bp. Signal maps and peak locations were used as input data to Active Motifs proprietary analysis program, which creates Excel tables containing detailed information on sample comparison, peak metrics, peak locations, and gene annotations.

**Statistical information**. Statistical analysis was performed with GraphPad Prism 8 software (GraphPad software, Inc.). All data are presented as mean ± standard error of the mean (SEM). The sample size for each experiment and the replicate number of experiments are included in the figure legends. Statistical analyses were performed using Student's $t$ test where applicable for comparison between two groups, and a one-way ANOVA test or two-way ANOVA was used for experiments involving more than two groups. Statistical significance was defined as $*P < 0.05$, $**P < 0.01$, $***P < 0.001$, $****P < 0.0001$; ns, not significant.

**Reporting summary**. Further information on research design is available in the Nature Research Reporting Summary linked to this article.

## Data availability

All RNA-seq and ChIP-seq data from this study were deposited in the Gene Expression Omnibus (GEO) with the accession number of GSE137126. The source data underlying all figures are provided as Source Data files. All other remaining data are available within the article and Supplemental Files, or available from the authors upon request.

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

## Acknowledgements

This work was supported by the Office of the Assistant Secretary of Defense for Health Affairs, through the Bone Marrow Failure Research Program—Idea Development Award under Award No. W81XWH-18-1–0265 to Y.L. Opinions, interpretations, conclusions, and recommendations are those of the author and are not necessarily endorsed by the Department of Defense. This work was also supported in part by R01HL150624, R56DK119524, R56AG05250, and a Leukemia & Lymphoma Society (LLS) Translational Research Program Grant 6581–20 to Y.L. S.C.N. was supported by a NIH F32 Award 1F32CA203049. The authors would like to acknowledge the Flow Cytometry Core and In vivo Therapeutic Core Laboratories, which were sponsored, in part, by the NIDDK Cooperative Center of Excellence in Hematology (CCEH) grant U54 DK106846. This work was supported, in part, by a Project Development Team within the ICTSI NIH/ NCRR Grant Number UL1TR001108. We would like to thank Dr. Yang Xu at USCD for providing the p53[R248W] mice and Dr. Daniel G Tenen at Harvard Medical School for providing the Cebpα plasmid to the study.

## Author contributions

S.C., Z.G., and Y.L. conceived the concept, designed the experiments, analyzed and interpreted the data, and wrote the manuscript. S.C., Q.W., H.Y., M.L.C., S.V., S.C.N., R.G., C.Y., M.K., Z.G., A.F., D.H., S.Z.L., S.B., W.C., E.R.W., B.R., Z.C., and N.L. performed the experiments. Q.W. performed the ChIP-seq data analysis. H.G. and Yunlong. L performed the RNA-seq data analysis. Y.S., T.N.W., D.C.L., H.S.B., L.D.M., G.H., R.K., M.C.Y., and H.E.B. provided reagents and constructive advice to the study.

## Competing interests

The authors declare no competing interests.
