## [Peer Review File · Nature Communications]

Reviewers' comments:

Reviewer #1 (Remarks to the Author):

The authors report that some p53 gain-of-function mutations, e.g. R248W, confers a competitive advantage to HSPCs following transplantation and promotes HSPC expansion after radiation-induced stress. This is linked to accumulation of H3K27me3 at gene promoters and repression of some genes regulating HSPC self-renewal and differentiation, e.g. Cebpa and Gadd45g. At least Gadd45g seems to be causative, since increased Gadd45g decreases the colony formation of p53 mutant BM cells. Mutant p53 interacts preferentially with EZH2 and genetic or pharmacologic inactivation of EZH2 reverses the effects of mutant p53. The authors work suggests an epigenetic mechanism by which mutant p53 drives clonal hematopoiesis. Overall, the data are good quality and the story is convincing. I have a few concerns that should be addressed.

1. Have the authors tested the importance of other repressed genes, such as Cebpa, for increased colony formation? Is increased expression of Cebpa also sufficient to bypass effects of mutant p53? Is inactivation of Gadd45g sufficient for increased HSPC self-renewal and colony formation?
2. The authors demonstrated the advantage of p53 mutant HSPCs in a battery of assays in Figure 2 and 3. However, only one Figure S3H appears to show the importance of Gadd45g. Can Gadd45g be tested in additional assays?
3. Does mutant p53 and EZH2 directly associate with Cebpa and Gadd45g target genes?
4. Figure 5e is not very convincing. Where is the bulk of Ezh2 in WT cells if not in the soluble nucleus or chromatin fractions?

Reviewer #2 (Remarks to the Author):

Chen et al. present a study aimed at understanding the role of mutant p53 in hematopoietic stem cells with the knowledge that mutations occur in clonal hematopoiesis of indeterminate potential (CHIP). By testing a selection of mutant p53 molecules in retroviral overexpression systems they show that some p53 mutations can result in increased serial replating activity and increased engraftment of transduced HSPCs in recipients. A knock-in mouse model (the p53 R248W mutation) is studied in more detail-HSCs heterozygous for the R248W mutation are shown to be more competitive in primary and secondary transplantation by approximately 3 1/2 fold. This competitiveness increases when engrafted recipients are irradiated, possibly due to reduced apoptosis of the mutant cells. To investigate the pathways through which this increased engraftment occurs, the authors perform genomic studies that rule out some previously-studied paradigms (mostly from epithelial cell/solid tumor studies) and instead focus on the in silico discovery of overlap between PRC2 (Ezh2)-regulated genes and mutant p53 regulated genes in HSPCs. A physical interaction between the p53 R248W mutant and Ezh2 is shown by IP and colocalization studies, and an overall increase in H3K27me3 is shown in p53 mutant HSPCs. ChIP for H3K27me3 shows an expansion of H3K27me3 overall, somewhat selectively on differentiation-associated genes. The role of enhanced Ezh2 activity is tested pharmacologically and genetically using engraftment and expansion of HSPCs, as well as limited gene expression to assess the requirement for Ezh2 in these phenotypes. These findings are very exciting given they reveal an elusive indirect effect of mutant p53 that has more to do with Ezh2 targets than p53 targets. This work is rigorously performed, particularly the quantification of HSC activities, and may bring to light a new, tissue-selective set of p53 functions and target genes relevant to the leukemia-predisposing CHIP condition with implications for other diseases with which CHIP is associated. However the connection between Ezh2 activity/its gene expression consequences and p53

mutation could be strengthened.

The specific concerns:

- 1) Chromatin immunoprecipitations studies are not performed quantitatively (with spike-in controls) raising some concerns regarding the interpretation of the re-distribution of H3K27me3 peaks. Can the authors show with a Venn diagram or table at least the number of called peaks in both WT and R248W lin-Kit⁺ replicates that overlap and that do not overlap with some examples of each beyond the 2 in Figure 4g (also the excel data described in the Methods)? Neither Figure 4e nor f really show the extent to which the same peaks are more enriched for H3K27me3 in the mutant cells or whether re-distribution of peaks truly occurs or both. Furthermore, the specific genes validated (CEBP α and GADD45 γ) by RT-PCR do represent more quantitative experiments, but the ChIP and transcript data are not matched for cell type (HSPC vs HSC). This makes it difficult to interpret the relationship between the gene expression observations and the histone modification observations.
- 2) The authors are unclear in Figure 4d what cell type is used for the quantification of H3K27me3 levels. In the methods, lineage-depleted HSPCs (kit⁺) are mentioned yet the legends say HSCs. This same lack of clarity of cell type is also true of Figure 5c-d: the legends say HSPC but the methods suggest that CD150/c-Kit are also used.
- 3) The localization of p53 mutant and Ezh2 is not very convincing at 40x with the Amnis approach. Can the authors also show confocal images with higher magnification/resolution?
- 4) How do the p53 mutant gene expression data compare with EED knockout LSK cells (published in 2014) or other PRC2-related data sets from HSPCs? These are likely more relevant GSEA comparisons than an ovarian cancer cell line (used in Figure 4b).

Typo issues:

- 1) The axis of the 6f graph goes to negative 20%. That seems un-necessary.
- 2) The heading of the first results section sounds like an aim of a grant rather than a heading.
- 3) line 168 has a word "deletion" which should probably not be there
- 4) line 190 the phrase clonal dominance should be clonally dominant

Reviewer #3 (Remarks to the Author):

Chen and colleagues investigated the role of missense p53 mutations in CHIP, a process where a large portion of mature hematopoietic cells derive from single stem progenitors. Using both overexpression and knock-in approaches, they determined that progenitor cells expressing mutant p53 had a competitive advantage, thus representing a larger percentage of cells in mature blood than controls. Further, radiation treatment conferred a strong selective advantage on mutant p53 expressing cells relative to cells with wild-type p53. Gene expression analysis in p53^{+/+} vs. p53^{R248W/+} HSPC suggested that EZH2 target genes are downregulated in mutant p53 expressing cells. This correlated with increased global levels of the EZH2-catalyzed H3K27me3 and specific increases in H3K27me3 at gene promoters. EZH2 and mutant p53 interact directly by co-immunoprecipitation. The authors then examined whether features of CHIP might be due to EZH2 activity in mutant p53 backgrounds. Both genetic and chemical inhibition of EZH2 in mutant p53 backgrounds reduced CHIP features, such as increased proportion of mutant p53-containing mature blood cells and restoration of certain EZH2 target genes.

The study is generally very well performed and a significant amount of the work points towards a clear role for mutant p53 and EZH2 in CHIP. The data are clearly presented, and the manuscript is well written. The manuscript has clear potential to provide both interesting insight into CHIP but also a potential therapeutic intervention point. Unfortunately, I have a few concerns at this time that reduce my enthusiasm for the work and bring into question the proposed model for p53 and EZH2 activity in CHIP.

The first concern involves what amount of the observations are due to a loss of normal p53 function relative to the activity of the missense p53 mutant (in most cases, R248W). Missense p53 mutations can serve two roles which are often difficult to tease apart. This includes a gain-of-function activity, as the authors describe, but also a dominant negative activity on the remaining wild-type p53 allele. Some results shown in Figure 2 (2F, 2G, and 2H) suggest loss of p53 (p53^{-/-}) and mutant p53 (p53^{R248W/+}) have similar phenotypes. This is discussed in the manuscript in lines 148-164.

Unfortunately, the authors leave behind this critical question for the remainder of the manuscript. The EZH2 transcriptional signature shown in Figure 4B and the increase in global levels of H3K27me3 shown in Figure 4C could be due to a loss of wild-type p53 activity instead of due to the presence of the R248W allele. This is a fairly significant discrepancy to resolve, in my opinion.

The second concern involves the connection between mutant p53 and EZH2, which forms part of the central thesis of the manuscript. The authors show EZH2 mRNA levels rise (but not protein) in the presence of p53 R248W and that EZH2 and p53 R248W physically interact. The authors suggest that the interaction between mutant p53 and EZH2 increases the fraction of EZH2 bound to chromatin (Figure 5E), but these data are fairly unconvincing, with almost no observable EZH2 bound to chromatin in WT cells (despite plenty of H3K27me3 in the genome). The conclusions would be stronger if one could show increased EZH2 occupancy on chromatin (and at the proposed genes) by chromatin IP (and doubly stronger if mutant p53 was also there).

Mutant p53 is incapable of binding to its normal binding sites and has been shown to be targeted by interactions with other transcription factors (ETS family, SREBP, etc). The increase in EZH2-dependent H3K27me3 appears to be fairly broad across genes, pointing towards an alternative EZH2/PRC targeting strategy (or just an increase in enzymatic activity but with normal targeting mechanisms). The alternative strategy is not discussed nor are potential known mechanisms of mutant p53 binding to chromatin (and how this might influence EZH2 targeting).

Single biological replicates of H3K27me3 ChIP-seq are generally insufficient given the key role these observations play in the proposed mechanism. While I do not think quantitative ChIP-seq experiments are necessarily required (using spike-in approaches), it's unclear that the data represent H3K27me3 or just background sheared chromatin. It would be more convincing to demonstrate these results are reproducible. Additionally, I would expect to see the genes with high H3K27me3 occupancy at the TSS to be significantly less expressed than those with low/no occupancy. This type of analysis would go a long way to demonstrating the general "success" of the experiment.

Additionally, the authors should discuss and investigate a key alternative hypothesis, which is the involvement H3K27me3 demethylases KMD6A/UTX and JMJD3. Decreased activity of those enzymes would phenocopy increased EZH2 activity. This is not discussed and represents an alternative mechanistic explanation for most of the observed data.

Reviewers' comments:

Reviewer #1 (Remarks to the Author):

The authors report that some p53 gain-of-function mutations, e.g. R248W, confers a competitive advantage to HSPCs following transplantation and promotes HSPC expansion after radiation-induced stress. This is linked to accumulation of H3K27me3 at gene promoters and repression of some genes regulating HSPC self-renewal and differentiation, e.g. *Cebpa* and *Gadd45g*. At least *Gadd45g* seems to be causative, since increased *Gadd45g* decreases the colony formation of p53 mutant BM cells. Mutant p53 interacts preferentially with EZH2 and genetic or pharmacologic inactivation of EZH2 reverses the effects of mutant p53. The authors work suggests an epigenetic mechanism by which mutant p53 drives clonal hematopoiesis. Overall, the data are good quality and the story is convincing. I have a few concerns that should be addressed.

1. Have the authors tested the importance of other repressed genes, such as *Cebpa*, for increased colony formation? Is increased expression of *Cebpa* also sufficient to bypass effects of mutant p53? Is inactivation of *Gadd45g* sufficient for increased HSPC self-renewal and colony formation?

Cebpa is required for myeloid differentiation and frequently disrupted in human AML. Loss of *Cebpa* enhances HSC self-renewal. We tested the importance of *Cebpa* on colony formation of p53 mutant BM cells and found that ectopic *Cebpa* expression decreased the colony formation of p53 mutant BM cells (New Supplementary Figure 5d). Thus, increased expression of *Cebpa* is sufficient to bypass effects of mutant p53 on colony formation. We showed that ectopic *Gadd45g* expression decreased the colony formation of p53 mutant BM cells (Supplementary Figure 5b). *Gadd45g*^{-/-} HSCs show enhanced reconstitution capability compared to WT HSCs in secondary transplantation assays, suggesting that loss of *Gadd45g* increases HSC self-renewal (reference #43). It is possible that inactivation of *Gadd45g* is responsible for increased self-renewal and colony formation seen in p53 mutant HSPCs.

2. The authors demonstrated the advantage of p53 mutant HSPCs in a battery of assays in Figure 2 and 3. However, only one Figure S3H appears to show the importance of *Gadd45g*. Can *Gadd45g* be tested in additional assays?

We introduced *Gadd45g* into wild-type and mutant p53 BM cells using retroviruses and transplanted transduced cells (GFP⁺) together with wild-type competitor BM cells into lethally-irradiated recipient mice. We found that ectopic *Gadd45g* expression decreases the engraftment of p53 mutant BM cells (New Supplementary Figure 5c), providing *in vivo* evidence that *Gadd45g* indeed is important for mutant HSPC function.

3. Does mutant p53 and EZH2 directly associate with *Cebpa* and *Gadd45g* target genes?

We performed p53 and EZH2 ChIP-qPCR assays in Lin⁻Kit⁺ bone marrow cells from p53^{+/+} and p53^{R248W/+} mice and found that both mutant p53 and EZH2 show increased association with *Cebpa* and *Gadd45g* (New Supplementary Figure 6e,f and data not shown). Both *Cebpa* and *Gadd45g* regulate many target genes in hematopoietic cells.

4. Figure 5e is not very convincing. Where is the bulk of Ezh2 in WT cells if not in the soluble nucleus or chromatin fractions?

We thank the reviewer for pointing out this discrepancy. We have repeated the fractionation experiments and included the cytosolic fraction, which has a substantial distribution of EZH2 (New Figure 5e). Our observations are consistent with previous reports showing the presence of EZH2 in cytosol.

References:

Gunawan, M. *et al.* The methyltransferase Ezh2 controls cell adhesion and migration through direct methylation of the extranuclear regulatory protein talin. *Nat Immunol.* 16(5):505-16 (2015).

Ougolkov, A.V. *et al.* Regulation of pancreatic tumor cell proliferation and chemoresistance by the histone methyltransferase enhancer of zeste homologue 2. *Clin Cancer Res.* 14(21):6790-6 (2008).

Reviewer #2 (Remarks to the Author):

Chen et al. present a study aimed at understanding the role of mutant p53 in hematopoietic stem cells with the knowledge that mutations occur in clonal hematopoiesis of indeterminate potential (CHIP). By testing a selection of mutant p53 molecules in retroviral overexpression systems they show that some p53 mutations can result in increased serial replating activity and increased engraftment of transduced HSPCs in recipients. A knock-in mouse model (the p53 R248W mutation) is studied in more detail-HSCs heterozygous for the R248W mutation are shown to be more competitive in primary and secondary transplantation by approximately 3 ½ fold. This competitiveness increases when engrafted recipients are irradiated, possibly due to reduced apoptosis of the mutant cells. To investigate the pathways through which this increased engraftment occurs, the authors perform genomic studies that rule out some previously-studied paradigms (mostly from epithelial cell/solid tumor studies) and instead focus on the in silico discovery of overlap between PRC2 (Ezh2)-regulated genes and mutant p53 regulated genes in HSPCs. A physical interaction between the p53 R248W mutant and Ezh2 is shown by IP and colocalization studies, and an overall increase in H3K27me3 is shown in p53 mutant HSPCs. ChIP for H3K27me3 shows an expansion of H3K27me3 overall, somewhat selectively on differentiation-associated genes. The role of enhanced Ezh2 activity is tested pharmacologically and genetically using engraftment and expansion of HSPCs, as well as limited gene expression to assess the requirement for Ezh2 in these phenotypes. These findings are very exciting given they reveal an elusive indirect effect of mutant p53 that has more to do with Ezh2 targets than p53 targets. This work is rigorously performed, particularly the quantification of HSC activities, and may bring to light a new, tissue-selective set of p53 functions and target genes relevant to the leukemia-predisposing CHIP condition with implications for other diseases with which CHIP is associated. However the connection between Ezh2 activity/its gene expression consequences and p53 mutation could be strengthened.

The specific concerns:

1) Chromatin immunoprecipitations studies are not performed quantitatively (with spike-in controls) raising some concerns regarding the interpretation of the re-distribution of H3K27me3 peaks.

Active Motif performed the ChIP-seq assays with spike-in controls. We forgot to state in the method. The spike-in adjustments were relatively minor and technical variations, including mapping rate, duplication rate, peak numbers, between the samples are low. Thus, we performed ChIP-seq studies quantitatively. Given that global signals are similar and the technical variation between the samples are low, we believe that the distribution of H3K27me3 peaks seen in p53 mutant HSPCs are real and reliable.

Can the authors show with a Venn diagram or table at least the number of called peaks in both WT and R248W lin-Kit⁺ replicates that overlap and that do not overlap with some examples of each beyond the 2 in Figure 4g (also the excel data described in the Methods)? Neither Figure 4e nor f really show the extent to which the same peaks are more enriched for H3K27me3 in the mutant cells or whether re-distribution of peaks truly occurs or both.

We have generated the Venn diagram as shown in new supplemental Figure 4a. We also provided additional representative loci for different categories of genes (new supplemental Figure 4b, c, and d).

As shown in the Venn diagram (new supplemental Figure 4a), 96.7% (2582 out of 2669) H3K27me3 peaks in WT cells also show H3K27me3 occupation in p53R248W mutant cells. Although more peaks are enriched in p53R248W cells using the same enrichment threshold, they are indeed associated with H3K27me3 in WT cells albeit lower enrichment. Taken together, we conclude that p53R248W mutant enhances H3K27me3 occupation rather than changes its genome-wide distribution in HSPCs. Therefore, we modified the old supplemental figure 3 by removing panels e and f, as well as modified the relevant text accordingly.

Furthermore, the specific genes validated (*CEBPα* and *GADD45γ*) by RT-PCR do represent more quantitative experiments, but the ChIP and transcript data are not matched for cell type (HSPC vs HSC). This makes it difficult to interpret the relationship between the gene expression observations and the histone modification observations.

Active Motif performed the ChIP-seq assays a few years ago. While the conventional ChIP-seq assays require millions of cells, we can only purify approximately 5,000 HSCs (CD48⁻CD150⁺LSKs) from each mouse. To overcome this technical limitation, we performed H3K27me3 ChIP-seq assays using Lin⁻Kit⁺ cells, which are HSPCs. We then confirmed ChIP-seq data in Lin⁻Kit⁺ cells using ChIP experiments (Fig. 4h). We agreed with the reviewer that it is difficult to interpret the relationship between the gene expression observations and the histone modification observations. We observed increased levels of H3K27me3 in p53 mutant HSCs (Fig. 4d). Further, we demonstrated that both *CEBPα* and *GADD45g* are downregulated in p53 mutant HSCs (Fig 4i, j). While we do not have H3K27me3 ChIP-seq data in HSCs, the expression of both *CEBPα* and *GADD45g* correlates with H3K27me3, suggesting that downregulation of *CEBPα* and *GADD45g* in HSCs is likely due to enrichment of H3K27me3 at their promoters.

2) The authors are unclear in Figure 4d what cell type is used for the quantification of H3K27me3 levels. In the methods, lineage-depleted HSPCs (kit+) are mentioned yet the legends say HSCs. This same lack of clarity of cell type is also true of Figure 5c-d: the legends say HSPC but the methods suggest that CD150/c-Kit are also used.

We provided more detailed information on flow cytometry analyses of HSCs according to reviewer's comments. For Figure 4d, lineage depleted HSPCs were stained with Sca1, Kit, and SLAM surface markers (CD48 and CD150) before fixation. We quantified the intensity of H3K27me3 in Lin⁻Sca1⁺Kit⁺CD48⁻CD150⁺ cells. For Figure 5c, 5d, S6c, and S6d, we quantified co-localization of EZH2 and p53 in Lin⁻Sca1⁺Kit⁺CD150⁺ cells. We specified the cell population in figure legends and methods.

3) The localization of p53 mutant and Ezh2 is not very convincing at 40x with the Amnis approach. Can the authors also show confocal images with higher magnification/resolution?

ImageStream technology allows for the combination of quantitative image analysis on the individual cell level with the ability to perform co-localization experiments on a statistically robust cell number making the ImageStream platform a more powerful tool than confocal microscopy. Unlike confocal microscopy, the data provided by the ImageStream is quantifiable and unbiased. Utilizing the software, we were able to calculate the similarity score (a log-transformed Pearson's correlation coefficient between the pixel values of two image pairs; in our case channel 2 and channel 3) which provides a measure of the degree of co-localization between p53 and Ezh2 staining without any input/bias from the user. We specifically looked at the similarity score within the nucleus by asking the software to draw a gate/mask region (as defined by DAPI staining; channel 7) of the individual cell being analyzed again without any user bias. Unlike confocal microscopy, the computer will do this on the gated population of our choice thus provided a better tool for analyzing small populations like hematopoietic stem cells with little to no manipulation prior to fixation. We gated on the lineage negative, Sca1 positive, cKit positive, CD150 positive (LSK CD150+) cells, just like one would when performing flow cytometry, then asked the software to provide a histogram for the entire sample (between 171-554 cells in each sample) of the similarity scores of each individual cell allowing one to see the variability of the sample. Then the software provides the geometric mean of the similarity score of the asked for gated region (the nucleus of the LSK CD150+ population) which we then averaged for three biological replicates. Therefore, even though confocal microscopy would provide a clear, higher resolution picture than the ImageStream (which we provided only as an example of the cells analyzed; see Figure 5c), the focus should be on the reproducible quantification of the p53 and Ezh2 co-localization within the nucleus of >171 LSK CD150+ cells per sample shown in Figure 5d.

For more information on this technology please see the following sources: IDEAS software user manual http://www.pedsresearch.org/uploads/pages/img/IDEAS_User_Manual_6.pdf; Maguire et al. (2011) Quantifying nuclear p65 as a parameter for NF-κB activation: correlation between ImageStream cytometry, microscopy and Western blot. *Cytometry A*. 79(6): 461-469.; George et al. (2006) Quantitative measurement of nuclear translocation events using similarity analysis of multispectral cellular images obtained in flow. *J. Immunol. Methods*. 311: 117-129; Henry et al. (2008) Quantitative image based apoptotic index

measurement using multispectral imaging flow cytometry: a comparison with standard photometric methods. *Apoptosis*. 13: 1054-1063.

4) How do the p53 mutant gene expression data compare with EED knockout LSK cells (published in 2014) or other PRC2-related data sets from HSPCs? These are likely more relevant GSEA comparisons than an ovarian cancer cell line (used in Figure 4b).

We agreed with the reviewer that using PRC2-related data sets from HSCs are more relevant than the one we used in the first submission. We performed GSEA analysis on two curated PRC2-related data sets from a Blood paper (Mochizuki-Kashio M, *Blood* 2015). The “H3K27me3 only genes” gene set includes genes that only marked by H3K27me3 in HSCs assessed by H3K27me3 ChIP-seq assays. The “EZH2 target_EZH1 no comp” data set includes genes that loss H3K27me3 level at least twofold after EZH2 deletion in LSKs (without EZH1 compensation) assessed by H3K27me3 ChIP-seq assays. As shown in New Figure 4b, GSEA analysis showed that genes that only marked by H3K27me3 were negatively enriched with significance in $p53^{R248W/+}$ mutant HSCs compared to that of the WT HSPCs. EZH2 target gene signature (without EZH1 compensation) was negatively enriched with significance in mutant HSCs compared to that of the WT HSPCs.

Reference:

Mochizuki-Kashio, M, *et al.* Ezh2 loss in hematopoietic stem cells predisposes mice to develop heterogeneous malignancies in an Ezh1-dependent manner. *Blood*. 126(10):1172-83 (2015).

Typo issues:

1) The axis of the 6f graph goes to negative 20%. That seems un-necessary.

We corrected Y-axis of Figure 6f.

2) The heading of the first results section sounds like an aim of a grant rather than a heading.

We changed the heading of the first result section to “Ectopic expression of *TP53* mutations identified in CHIP enhances HSPC repopulating potential”.

3) line 168 has a word “deletion” which should probably not be there

We corrected that in the manuscript.

4) line 190 the phrase clonal dominance should be clonally dominant.

We changed clonal dominance to clonally dominant in revised manuscript.

Reviewer #3 (Remarks to the Author):

Chen and colleagues investigated the role of missense p53 mutations in CHIP, a process

where a large portion of mature hematopoietic cells derive from single stem progenitors. Using both overexpression and knock-in approaches, they determined that progenitor cells expressing mutant p53 had a competitive advantage, thus representing a larger percentage of cells in mature blood than controls. Further, radiation treatment conferred a strong selective advantage on mutant p53 expressing cells relative to cells with wild-type p53. Gene expression analysis in p53^{+/+} vs. p53^{R248W/+} HSPC suggested that EZH2 target genes are downregulated in mutant p53 expressing cells. This correlated with increased global levels of the EZH2-catalyzed H3K27me3 and specific increases in H3K27me3 at gene promoters. EZH2 and mutant p53 interact directly by co-immunoprecipitation. The authors then examined whether features of CHIP might be due to EZH2 activity in mutant p53 backgrounds. Both genetic and chemical inhibition of EZH2 in mutant p53 backgrounds reduced CHIP features, such as increased proportion of mutant p53-containing mature blood cells and restoration of certain EZH2 target genes.

The study is generally very well performed and a significant amount of the work points towards a clear role for mutant p53 and EZH2 in CHIP. The data are clearly presented, and the manuscript is well written. The manuscript has clear potential to provide both interesting insight into CHIP but also a potential therapeutic intervention point. Unfortunately, I have a few concerns at this time that reduce my enthusiasm for the work and bring into question the proposed model for p53 and EZH2 activity in CHIP.

The first concern involves what amount of the observations are due to a loss of normal p53 function relative to the activity of the missense p53 mutant (in most cases, R248W). Missense p53 mutations can serve two roles which are often difficult to tease apart. This includes a gain-of-function activity, as the authors describe, but also a dominant negative activity on the remaining wild-type p53 allele. Some results shown in Figure 2 (2F, 2G, and 2H) suggest loss of p53 (p53^{-/-}) and mutant p53 (p53^{R248W/+}) have similar phenotypes. This is discussed in the manuscript in lines 148-164. Unfortunately, the authors leave behind this critical question for the remainder of the manuscript. The EZH2 transcriptional signature shown in Figure 4B and the increase in global levels of H3K27me3 shown in Figure 4C could be due to a loss of wild-type p53 activity instead of due to the presence of the R248W allele. This is a fairly significant discrepancy to resolve, in my opinion.

Based on Reviewer's comment, we performed GSEA analysis on curated PRC2-related data sets from HSPCs (Mochizuki-Kashio M, Blood 2015). As shown in New Figure 4b, GSEA analysis showed that genes that only marked by H3K27me3 were negatively enriched with significance in mutant HSCs compared to that of the WT HSPCs. EZH2 target gene signature (without EZH1 compensation) was negatively enriched with significance in mutant HSCs compared to that of the WT HSPCs.

Wild type p53 has not been shown to be associated with EZH2 activity or H3K27me3 in the literature. We performed GSEA analysis on RNA-seq data of p53^{+/+} and p53^{-/-} HSPCs and found that EZH2 target gene signature was not significantly changed in p53^{-/-} HSPCs compared to WT HSPCs (new Supplemental Figure 3e), which is different from what we observed in p53 mutant LSKs (New Fig. 4b), suggesting that loss of wild-type p53 may affect neither EZH2 activity nor H3K27me3 in HSPCs. Ectopic expression of mutant p53, but not wild-type p53, enhances H3K27me3 in 32D cells (Fig. 5a). Further, we found that several mutant p53 proteins show enhanced association with EZH2 compared to wild-type p53 (Fig. 5b). The expression of p53 target genes in HSCs, including p21 and Necdin, was

comparable in p53^{+/+} and p53^{R248W/+} HSCs (Fig. 4a), suggesting that p53R248W mutant does not have dominant negative activity in HSCs. While both loss of p53 (p53^{-/-}) and mutant p53 (p53R248W/+) enhances HSC repopulating potential (Fig. 2f), our data suggest that increased levels of H3K27me3 in p53 mutant HSPCs (Figure 4C) is likely due to the presence of the R248W allele, but not the result of losing wild-type p53 activity. We discussed these in revised manuscript.

Reference:

Mochizuki-Kashio, M, et al. Ezh2 loss in hematopoietic stem cells predisposes mice to develop heterogeneous malignancies in an Ezh1-dependent manner. *Blood*. 126(10):1172-83 (2015).

The second concern involves the connection between mutant p53 and EZH2, which forms part of the central thesis of the manuscript. The authors show EZH2 mRNA levels rise (but not protein) in the presence of p53 R248W and that EZH2 and p53 R248W physically interact. The authors suggest that the interaction between mutant p53 and EZH2 increases the fraction of EZH2 bound to chromatin (Figure 5E), but these data are fairly unconvincing, with almost no observable EZH2 bound to chromatin in WT cells (despite plenty of H3K27me3 in the genome). The conclusions would be stronger if one could show increased EZH2 occupancy on chromatin (and at the proposed genes) by chromatin IP (and doubly stronger if mutant p53 was also there).

We thank the reviewer for pointing out this discrepancy. We have repeated the fractionation assays and included the cytosolic fraction, which has a substantial distribution of EZH2 (new Figure 5e). Our observations are consistent with previous reports that showed the presence of EZH2 in cytosol.

Reference:

Gunawan, M. *et al.* The methyltransferase Ezh2 controls cell adhesion and migration through direct methylation of the extranuclear regulatory protein talin. *Nat Immunol*. 16(5):505-16 (2015).

Ougolkov, A.V. *et al.* Regulation of pancreatic tumor cell proliferation and chemoresistance by the histone methyltransferase enhancer of zeste homologue 2. *Clin Cancer Res*. 14(21):6790-6 (2008).

Mutant p53 is incapable of binding to its normal binding sites and has been shown to be targeted by interactions with other transcription factors (ETS family, SREBP, etc). The increase in EZH2-dependent H3K27me3 appears to be fairly broad across genes, pointing towards an alternative EZH2/PRC targeting strategy (or just an increase in enzymatic activity but with normal targeting mechanisms). The alternative strategy is not discussed nor are potential known mechanisms of mutant p53 binding to chromatin (and how this might influence EZH2 targeting).

As shown in the Venn diagram (new supplemental Figure 4a), 96.7% H3K27me3 peaks in WT cells also show H3K27me3 occupation in p53R248W mutant cells. Although more peaks are enriched in p53R248W cells using the same enrichment threshold, they are indeed associated with H3K27me3 in WT cells albeit lower enrichment. Taken together, we conclude

that p53^{R248W} mutant enhances H3K27me3 occupation rather than changes its genome-wide distribution in HSPCs. We found that mutant p53 interacts with EZH2 (Fig. 5b, c) and enhances its association with the chromatin in HSPCs (Fig. 5e), thereby increasing H3K27me3 occupancy at genes important for HSC self-renewal and differentiation. We discussed this in revised manuscript.

Single biological replicates of H3K27me3 ChIP-seq are generally insufficient given the key role these observations play in the proposed mechanism. While I do not think quantitative ChIP-seq experiments are necessarily required (using spike-in approaches), it's unclear that the data represent H3K27me3 or just background sheared chromatin. It would be more convincing to demonstrate these results are reproducible.

We did perform the ChIP-seq assays with spike-in controls, see revised Method. Both wild-type and p53 mutant HSPCs are pooled from several age and sex-matched mice for ChIP-seq analysis. We have analyzed previously published H3K27me3 ChIP-seq in HSPCs. We found that more than 80% of peaks in their study are shared by our own dataset in the manuscript (Figure R1, see below). We also confirmed ChIP-seq data with ChIP RT-PCR assays. Taken together, we believe that peaks identified from our ChIP-seq analysis represent real H3K27me3 enriched genomic regions.

References:

Collins, C, *et al.* C/EBP α is an essential collaborator in Hoxa9/Meis1-mediated leukemogenesis. *Proc Natl Acad Sci U S A.* 111(27):9899-904 (2014).

Additionally, I would expect to see the genes with high H3K27me3 occupancy at the TSS to be significantly less expressed than those with low/no occupancy. This type of analysis would go a long way to demonstrating the general "success" of the experiment.

We performed comprehensive analysis of H3K27me3 ChIP-seq data with RNA-seq data from p53^{+/+} and p53^{R248W/+} HSPCs. Our analysis showed expected lower expression of genes targeted by H3K27me3 identified from our ChIP-seq study, as shown in Figure R2.

Additionally, the authors should discuss and investigate a key alternative hypothesis, which is the involvement H3K27me3 demethylases KMD6A/UTX and JMJD3. Decreased activity of those enzymes would phenocopy increased EZH2 activity. This is not discussed and represents an alternative mechanistic explanation for most of the observed data.

We agree with reviewer that decreased activity of H3K27me3 demethylases, such as KMD6A/UTX and JMJD3, may be an alternative mechanism leading to increased EZH2 activity. We discussed this point in revised manuscript.

Figure R1. Venn diagram shows the overlap of H3K27me3 peaks in HSPCs from our and previous CHIP-seq analysis.

Figure R2. Whisker box plot shows average expression of genes targeted by H3K27me3 are generally repressed. H3K27me3+, genes targeted by H3K27me3 identified by our CHIP-seq in p53 WT HSPCs. H3K27me3-, non-H3K27me3 target genes.

REVIEWERS' COMMENTS:

Reviewer #2 (Remarks to the Author):

This revised study focuses on a new and unexpected finding linking EZH2 activity to mutant p53 specifically. The gene expression and molecular functional data provide strong evidence for this new mechanism and its impact on HSPC competition with surrounding WT cells. However in parts of the text, there is also an implication that dominant interfering activity is not a mechanism that occurs with these mutants in this cell type. The similar level of p53 target gene expression as shown in Figure 4a (and GEO data) and lack of comparison between R248W/+ to R248W/R248W cells or assessment of p53-regulated genes in an inducible setting (eg upon DNA damage) makes me concerned that dominant interfering activity may well be part of the biological phenotypes shown, rather than exclusively the gain of function activity as implicated by the writing. This seems to be a concern shared by reviewer 3. In fact, the revised manuscript now shows that C/EBPa and GADD45g represent EZH2 targets that exhibit increased H3K27me3 in the presence of the R248W mutation and that TPO cannot induce GADD45g as well in the presence of this mutant. Even this figure could be interpreted as the p53 mutant exhibiting dominant interfering activity, if GADD45g is a p53 target in an inducible setting (DNA damage) and by TPO signaling, which I am not sure is true or not (however, one of the co-authors has published on TPO/p53 signaling). The data cited to support the genetic proof of GOF activity (refs. 29-31) represent studies performed with sufficiently different cell contexts to make these data not helpful to extrapolate to the current manuscript.

In summary the authors discover a new surprising mechanism for p53 mutant function that might be very important in the clinical management and treatment of preleukemia and AML as well as potentially other cancers. The findings shown here will be of significant interest to a wide variety of readers and there are many methods used to confirm the basic findings. However, the authors should adjust the text (possible locations: title, abstract, discussion) to accommodate the significant evidence that mutant p53 frequently acts as a dominant interfering molecule as well.

Minor comments

1) the final sentence in line 470 does not seem relevant to this study and could be instead replaced by an ending to the prior sentence like "...self-renewal, providing a model for the dissection of mutant p53 in leukemia progression..." or something of this nature

2) can the authors comment on kdm6a/jmjd3 transcript levels in their RNAseq data? (new comments on line 419)

Reviewer #3 (Remarks to the Author):

The authors took significant time and effort to respond to the few issues that were raised in the previous review. This includes additional experimental evidence, synthesis of prior literature, and additional clarification/discussion in the text.

I feel the manuscript is now quite strong and that my prior concerns were adequately addressed in the review process.

REVIEWERS' COMMENTS:

Reviewer #2 (Remarks to the Author):

This revised study focuses on a new and unexpected finding linking EZH2 activity to mutant p53 specifically. The gene expression and molecular functional data provide strong evidence for this new mechanism and its impact on HSPC competition with surrounding WT cells. However in parts of the text, there is also an implication that dominant interfering activity is not a mechanism that occurs with these mutants in this cell type. The similar level of p53 target gene expression as shown in Figure 4a (and GEO data) and lack of comparison between R248W/+ to R248W/R248W cells or assessment of p53-regulated genes in an inducible setting (eg upon DNA damage) makes me concerned that dominant interfering activity may well be part of the biological phenotypes shown, rather than exclusively the gain of function activity as implicated by the writing. This seems to be a concern shared by reviewer 3. In fact, the revised manuscript now shows that C/EBPa and GADD45g represent EZH2 targets that exhibit increased H3K27me3 in the presence of the R248W mutation and that TPO cannot induce GADD45g as well in the presence of this mutant. Even this figure could be interpreted as the p53 mutant exhibiting dominant interfering activity, if GADD45g is a p53 target in an inducible setting (DNA damage) and by TPO signaling, which I am not sure is true or not (however, one of the co-authors has published on TPO/p53 signaling).

TP53 mutations have been shown to have Loss-of-Function (LOF), Dominant-Negative (DN) and Gain-of-Function (GOF) properties in human cancer. Our goal is not to determine whether mutant p53 has LOF, DN and/or GOF properties, but rather to investigate how mutant p53 drive clonal hematopoiesis. We did compare gene expression between $p53^{R248W/+}$ and $p53^{R248W/R248W}$ HSCs and found that the expression of p53 target genes, including *p21* and *Necdin*, are comparable between $p53^{R248W/+}$ and $p53^{R248W/R248W}$ HSCs (Figure 1). We agreed with the reviewer that dominant interfering activity may be part of the biological phenotypes shown and have changed the statement in the discussion (see below).

The data cited to support the genetic proof of GOF activity (refs. 29-31) represent studies performed with sufficiently different cell contexts to make these data not helpful to extrapolate to the current manuscript.

We agreed with the reviewer that genetic proof of GOF activity (refs. 29-31) represent studies performed with different cell contexts. We now included a recent paper from the Scott Lowe

laboratory showing that Gain-of-Function mutant p53 plays an important role in myeloid leukemia [New Reference #30: Loizou, E. *et al.* A Gain-of-Function p53-Mutant Oncogene Promotes Cell Fate Plasticity and Myeloid Leukemia through the Pluripotency Factor FOXH1. *Cancer Discov.* **9**,962-979 (2019)].

In summary the authors discover a new surprising mechanism for p53 mutant function that might be very important in the clinical management and treatment of preleukemia and AML as well as potentially other cancers. The findings shown here will be of significant interest to a wide variety of readers and there are many methods used to confirm the basic findings. However, the authors should adjust the text (possible locations: title, abstract, discussion) to accommodate the significant evidence that mutant p53 frequently acts as a dominant interfering molecule as well.

Dysregulated epigenetic pathways has been implicated in mutant p53-driven tumorigenesis [Reference #36: Zhu, J. *et al.* Gain-of-function p53 mutants co-opt chromatin pathways to drive cancer growth. *Nature* **525**, 206-11 (2015)]. We found that mutant p53 interacts with EZH2 and enhances H3K27me3 in HSPCs. We did not state that mutant p53 has GOF property in title, abstract, or conclusion, but only stated that mutant p53 could have GOF property in the discussion section. We agreed with the reviewer that mutant p53 frequently acts as a dominant interfering molecule and have cited a recent paper [New reference #51, Boettcher, S. *et al.* A dominant-negative effect drives selection of *TP53* missense mutations in myeloid malignancies. *Science* **365**,599-604 (2019)]. We have made the following statement in revised manuscript “While a dominant-negative (DN) effect has been shown to drive selection of *TP53* missense mutations in myeloid malignancies⁵¹, GOF mutant p53 appears to play an important role in myeloid leukemia³⁰. Our work suggests that both DN and GOF properties may contribute to enhanced HSC self-renewal seen in *p53*^{R248W/+} mice.”

Minor comments

1) the final sentence in line 470 does not seem relevant to this study and could be instead replaced by an ending to the prior sentence like “...self-renewal, providing a model for the dissection of mutant p53 in leukemia progression...” or something of this nature

We deleted the final sentence from the manuscript.

can the authors comment on *kdm6a/jmjd3* transcript levels in their RNAseq data? (new comments on line 419)
Our RNA-seq data revealed that the expression of

2)

Kdm6a and *Jmjd3* are comparable between *p53*^{+/+} and *p53*^{R248W/+} HSPCs (Figure 2). Thus, *Kdm6a* and *Jmjd3* may not involve in regulating H3K27me3 in p53 mutant HSPCs.

Reviewer #3 (Remarks to the Author):

The authors took significant time and effort to respond to the few issues that were raised in the previous review. This includes additional experimental evidence, synthesis of prior literature, and additional clarification/discussion in the text.

I feel the manuscript is now quite strong and that my prior concerns were adequately addressed in the review process.

We are glad to know that reviewer #3 is satisfied with the revised manuscript.